# Leveraging a gain-of-function allele of *Caenorhabditis elegans paqr-1* to elucidate membrane homeostasis by PAQR proteins

**Kiran Busayavalasa**[1], **Mario Ruiz**[1], **Ranjan Devkota**[1], **Marcus Ståhlman**[2], **Rakesh Bodhicharla**[1], **Emma Svensk**[1], **Nils-Olov Hermansson**[3], **Jan Borén**[2], **Marc Pilon**[1] *

**1** Department of Chemistry and Molecular Biology, University of Gothenburg, Gothenburg, Sweden,
**2** Department of Molecular and Clinical Medicine/Wallenberg Laboratory, Institute of Medicine, University of Gothenburg, Gothenburg, Sweden, **3** Discovery Biology, Discovery Sciences, R&D, AstraZeneca, Gothenburg, Sweden

* marc.pilon@cmb.gu.se

**Data Availability Statement:** All relevant data are within the manuscript and its Supporting Information files.

## Abstract

The *C. elegans* proteins PAQR-2 (a homolog of the human seven-transmembrane domain AdipoR1 and AdipoR2 proteins) and IGLR-2 (a homolog of the mammalian LRIG proteins characterized by a single transmembrane domain and the presence of immunoglobulin domains and leucine-rich repeats in their extracellular portion) form a complex that protects against plasma membrane rigidification by promoting the expression of fatty acid desaturases and the incorporation of polyunsaturated fatty acids into phospholipids, hence increasing membrane fluidity. In the present study, we leveraged a novel gain-of-function allele of PAQR-1, a PAQR-2 paralog, to carry out structure-function studies. We found that the transmembrane domains of PAQR-2 are responsible for its functional requirement for IGLR-2, that PAQR-1 does not require IGLR-2 but acts via the same pathway as PAQR-2, and that the divergent N-terminal cytoplasmic domains of the PAQR-1 and PAQR-2 proteins serve a regulatory function and may regulate access to the catalytic site of these proteins. We also show that overexpression of human AdipoR1 or AdipoR2 alone is sufficient to confer increased palmitic acid resistance in HEK293 cells, and thus act in a manner analogous to the PAQR-1 gain-of-function allele.

## Author summary

Cells are enclosed within membranes primarily composed of fat. When membranes contain much saturated fats, they tend to become more rigid, as with butter. Conversely, when membranes are rich in unsaturated fats, they become more fluid, as with vegetable oils. Our goal is to better understand how cells monitor and adjust the composition and properties of their membranes. We focus on a small group of proteins found in all animals, and called AdipoR1 and AdipoR2 in humans, and PAQR-1 and PAQR-2 in the worm *Caenorhabditis elegans*. We now found a version of PAQR-1 that is more "active", and promotes increased levels of unsaturated fats in membranes. By swapping different

**Funding:** This work was funded by Vetenskapsrådet (Dnr: 2016-03676); www.vr.se), Cancerfonden (Dnr 16 0693; www.cancerfonden. se), Carl Trygger Stiftelsen (CTS 16:365; www. carltryggerstiftelse.se), Diabetesfonden (DIA2016-109; www.diabetes.se/diabetesfonden), Kungliga Vetenskaps och Vitterhets-Samhället (www.kvvs. se), Åke Wibergs stiftelse (M19-0256; www.ake-wiberg.se) and Wilhelm och Martina Lundgrens Stiftelse (www.wmlundgren.se). The funders had no role in study design, data collection and analysis, decision to publish, or preparation of the manuscript.

**Competing interests:** The authors have declared that no competing interests exist.

parts of the PAQR-1 protein with those of PAQR-2, we were able to determine which protein parts played which roles. We found that it is the transmembrane domains of PAQR-2 that dictate its requirements for another protein called IGLR-2 and that the intracellular domains of PAQR-1 and PAQR-2 play a regulatory role. These studies help understand how AdipoR1 and AdipoR2 regulate membrane composition in human cells, which is a vital function for us to thrive on diets that vary greatly in the types of fats that they contain.

## Introduction

Maintenance of cell membrane homeostasis relies on regulatory proteins that sense and respond to properties such as lipid composition, thickness, compressibility, lateral mobility and curvature [1, 2]. The best understood example is perhaps the bacterial multi-pass protein DesK that undergoes a conformational change in response to increased membrane rigidification, resulting in the activation of its kinase domain [3–6]. Similarly, the yeast single-pass plasma membrane protein Mga2 rotates along its long axis when the surrounding acyl chains are densely packed, also resulting in its activation [7]. In eukaryotes, the SREBPs are regulated by the availability of specific lipids such as cholesterol in the ER membrane [8], PCYT1A is activated by association with packing defects in the inner nuclear membrane [9–11] and IRE1 is activated by multimerization in response to membrane thickening in the ER [1, 12]. Recently we identified a novel regulator of membrane homeostasis in animal cells, namely the PAQR-2/IGLR-2 complex in the nematode *C. elegans* [13–18]. The present work helps define the structure-functional basis of fluidity sensing by this complex.

The PAQR family of proteins (named after the founding members Progestin and AdipoQ Receptors) are characterized by the presence of seven transmembrane domains oriented such that their N-terminus is cytosolic [19]. PAQR proteins all likely act as hydrolases but with a wide range of substrates and biological functions [20]. The crystal structures of AdipoR1 and AdipoR2, which are PAQR proteins that may act as receptors for adiponectin, have been partially resolved and show that the transmembrane domains form a barrel with a cavity that opens on the cytosolic side where it may allow the in/out flow of substrates and products of a hydrolytic reaction coordinated by a zinc ion [21, 22]. There are yeast homologs to the AdipoRs that act as ceramidases signaling through sphingolipids to regulate lipid/membrane homeostasis [23–26], and a ceramidase activity has also been documented for the human AdipoRs [22, 27–29], though other substrates may exist. The structures solved so far do not include the cytoplasmic N-terminal domain, which contains most of the sequence divergence among AdipoR-like proteins, and the function of this domain therefore remains unknown.

In *C. elegans*, there are five PAQR proteins, including the AdipoR homologs PAQR-1 and PAQR-2, with PAQR-2 being best characterized [30]. We previously showed that PAQR-2 is a regulator of membrane fluidity that acts together with its dedicated partner, the single-pass transmembrane protein IGLR-2 [16, 17]. *paqr-2* and *iglr-2* single and double mutants have the same phenotypes, including a characteristic tail tip defect and intolerance to cold and dietary saturated fatty acids (SFAs) [16, 31]. Additionally, the *paqr-2* mutant, and presumably the *iglr-2* mutant as well, also exhibits several phenotypes secondary to its primary membrane homeostasis defect, including defects in lifespan [30], vitellogenin trafficking [15], brood size [30], locomotion [30], autophagy [32] and proteostasis [33]. These phenotypes are secondary to the primary membrane fluidity defects of *paqr-2* and *iglr-2* mutants because they can be suppressed fully or partially by low, fluidizing concentrations of mild detergents [18], by providing

supplements of unsaturated fatty acids [16, 18], or by secondary mutations that increase the relative abundance of unsaturated fatty acids (UFAs) among phospholipids, such as *mdt-15 (et14)*, *nhr-49(et8)*, *fld-1(et48)* and several others [14, 17, 18]. PAQR-2 and IGLR-2 mostly act cell non-autonomously: expression in one large tissue (e.g. gonad sheath cells, intestine or hypodermis) is sufficient to rescue the entire worm; the one exception is the tail tip which requires local expression likely because it constitutes a site of poor lipid exchange with the rest of the worm [15]. Additionally, bifluorescence complementation (BiFC) showed that the proteins PAQR-2 and IGLR-2 interact with each other on the plasma membrane, and GFP reporters showed that they are both strongly expressed in the gonad sheath cells and weakly in many other tissues [17].

Less is known about the function or mechanism of action of PAQR-1, though one study indicated that it may modulate lipid metabolism and ER stress [34]. Recently, and to identify additional components of the *paqr-2* pathway, we performed a forward genetic screen to isolate mutations that enhance the ability of *mdt-15(et14)* to suppress the SFA intolerance of the *paqr-2(tm3410)* null mutant. Of 15 novel mutants isolated in that screen, eight were alleles of the gene *fld-1* that functions by limiting the levels of phospholipids containing long chain polyunsaturated fatty acids (PUFAs); *fld-1* mutants therefore have elevated PUFA levels among their phospholipids [14]. Another mutation was a loss-of-function allele of the acyl-CoA synthetase *acs-13*; this mutation also promotes the incorporation of PUFAs into phospholipids [35]. In the present work, one of the remaining suppressor mutations has now been identified as a gain-of-function (*gof*) allele of *paqr-1*, i.e. *paqr-1(et52)*. This is interesting because while the *paqr-1* single mutant has no obvious phenotype, the double mutant *paqr-1;paqr-2* shows much more severe defects than either single mutants, suggesting functional overlap between the two genes [30]. Here, we show that the *paqr-1(et52)* allele acts through the same pathway as *paqr-2*. We also show that *paqr-1(et52)* suppresses the phenotypes of *paqr-2* and *paqr-2 iglr-2* double mutants better than it suppresses the *iglr-2* single mutant, which suggests that the PAQR-2 protein inhibits the *gof* PAQR-1 only when the IGLR-2 protein is also absent. Finally, we used domain-swapping experiments to show that the distinct intracellular N-terminal domains serve regulatory functions in the PAQR-1 and PAQR-2 proteins. This study is an important advance in our understanding of how PAQR-type proteins are regulated to achieve membrane homeostasis.

## Results

### *paqr-1(et52)* is a *gof* allele

The *paqr-1(et52)* allele was isolated in a previously published screen for suppressors of the *paqr-2(tm3410)* mutant SFA intolerance phenotype [14]. In this screen, the SFA-rich diet is achieved by cultivating *C. elegans* on plates containing 20 mM glucose, which is converted to SFAs by the dietary *E. coli*, also as previously described [16]. The amino acid sequences of PAQR-1 and PAQR-2 proteins are highly conserved throughout the 7 transmembrane domains and membrane-proximal ~100 amino acids of the cytoplasmic domain, but show very little sequence homology within the remainder of the large N-terminal regions, i.e. the first 200 amino acids of PAQR-1 and the first 309 amino acids of PAQR-2 (**Fig 1A**). The *paqr-1(et52)* mutation causes an R109C amino acid substitution about halfway within the cytoplasmic N-terminal domain, which suggests a regulatory function for this non-conserved domain (**Fig 1A**). The levels of protein expression appear mostly unaffected by the R109C amino acid substitution: Western blots against HA-tagged proteins expressed from CRISPR/Cas9-modified endogenous wild-type *paqr-1* or mutant *paqr-1(et52)* loci show similar expression levels throughout development for both allele (both are most abundant in embryos and

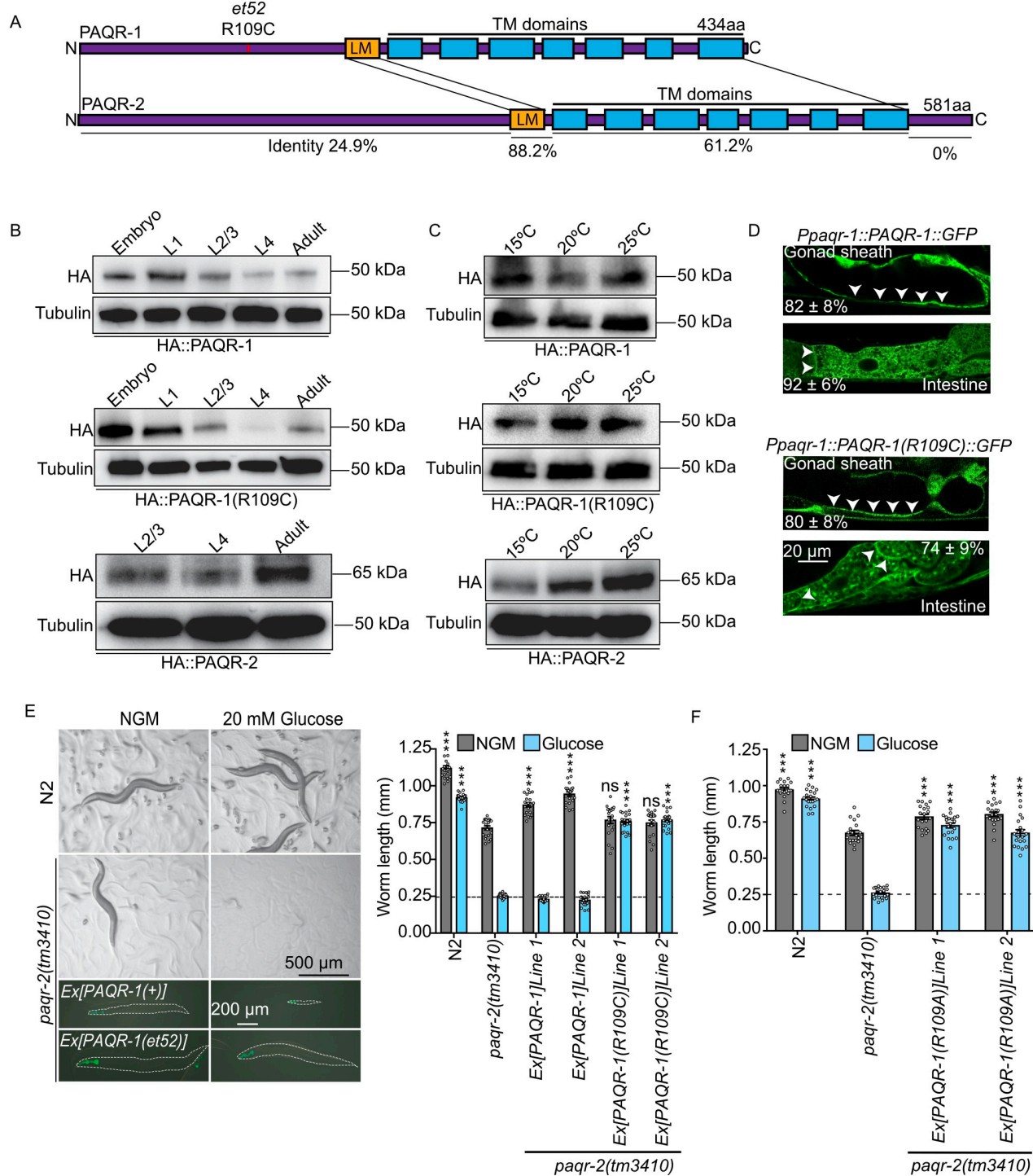

**Fig 1. *paqr-1(et52)* is a *gof* allele and an R109A substitution also acts as a *gof paqr-1* allele. (A)** Cartoon representation of the PAQR-1 and PAQR-2 proteins indicating the transmembrane domains (TM), localization motif (LM) and the position of the R109C substitution encoded by the *et52* allele in the cytoplasmic N terminus. The percentages of sequence identity between the various domains of PAQR-1 and PAQR-2 are indicated. **(B-C)** Western blot detection of HA-tagged proteins expressed from the CRISPR/Cas9-modified endogenous *paqr-1(syb1401)*, *paqr-1(syb364)* and *paqr-2(syb1401)* alleles now encoding HA::PAQR-1, HA::PAQR-1(R109C) and HA::PAQR-2, respectively. **(D)** Confocal images of transgenic adult *C. elegans* expressing *Ppaqr-1::PAQR-1WT::GFP* and *Ppaqr-1::PAQR-1R109C::GFP*, with frequencies of transgenic worms expressing the reporter in gonad sheath cells or intestine indicated (n = 50). Arrowheads indicate enrichment of the reporter on plasma membranes. Expression in body muscle and other tissues was also observed but at much lower frequencies. **(E)** The glucose intolerance of *paqr-2* mutant worms is abrogated by *PAQR-1(R109C)* but not wild-type *PAQR-1* when provided as transgenes. Transgenic worms were identified by the presence of the GFP transformation marker *myo-2::*

*GFP*, and the outlines of worms are indicated by dashed lines in the left panels. The right-side graph shows worm length 72 hours after placing L1s on normal plates or plates containing 20 mM glucose. **(F)** Expression of *PAQR-1(R109A)* also abrogates the glucose intolerance phenotype of *paqr-2* mutant worms; the graph shows the length of worms placed as L1s on the indicated media and measured 72 hours later. The dashed lines in E and F indicate the approximate size of the L1 larvae at the start of the experiment. Significant differences compared to the *paqr-2* genotype are indicated where: $^{*}$p<0.05, $^{**}$p<0.01 and $^{***}$p<0.001 (ns: not significant).

L1s; **Fig 1B**) or when cultivated at various temperatures (**Fig 1C**). In contrast, PAQR-2 expression is highest in adult and increases with temperature (**Fig 1B and 1C**). The localization of GFP translational reporters is also unchanged between the wild-type PAQR-1 and the mutant PAQR-1(R109C) proteins, which are both expressed in several tissues, but predominantly in the intestine and gonad sheath cells, which is the site of strongest PAQR-2 expression [17] (**Fig 1D**). In spite of unchanged expression levels and localization, *paqr-1(et52)* is clearly a *gof* mutation because providing it as a multicopy transgene in a *paqr-2* mutant background, where the endogenous *paqr-1* is wild-type, efficiently suppresses the SFA intolerance, cold intolerance and tail tip defect of the *paqr-2* mutant (**Fig 1E**; **S1A–S1C Fig**). In contrast, providing wild-type *paqr-1* as a multicopy transgene efficiently suppresses only the cold intolerance phenotype of the *paqr-2* mutant, partially suppresses the tail tip defect and does not suppress at all the intolerance to SFAs, which is a harder challenge for the *paqr-2* mutant since many *paqr-2* suppressor alleles that fully rescue growth at 15˚C do not effectively rescue growth on glucose [14, 18, 35](**Fig 1E**; **S1A–S1C Fig**). These results suggest that the R109C amino acid substitution endows the PAQR-1 protein with augmented activity compared to the wild-type protein, i.e. that it is a *gof* allele.

## PAQR-1(R109A) also acts as a gain-of-function allele

The *paqr-1(et52)* allele corresponds to a R109C substitution that could promote dimerization via disulfide bridges involving the cysteine. This is however not the case: substituting the arginine at position 109 by an alanine (R109A) also results in an equally potent *paqr-1 gof* allele that efficiently suppresses the glucose and cold intolerance and tail tip defects of the *paqr-2* mutant (**Fig 1F** and **S1D and S1E Fig**). We conclude that dimerization via disulfide bridges does not explain the *gof* nature of the R109C allele, and that the bulky and polarized arginine at position 109 limits the activity of the wild-type PAQR-1, perhaps by imposing an inhibitory structure or facilitating an inhibitory interaction that is abolished when the arginine is replaced by another amino acid.

## *paqr-1(et52)* suppresses *paqr-2* mutant phenotypes

Having established, using transgenic animals, that *paqr-1(et52)* is a *gof* allele, we proceeded to its more detailed characterization. The *paqr-1(et52)* mutation suppresses, partially or entirely, the defects in pharyngeal pumping rate, brood size, lifespan, defecation rate, and locomotion rate of the *paqr-2* mutant (**S1F–S1J Fig**). Indeed, the *paqr-1(et52)* mutation partially or entirely suppresses all *paqr-2* defects tested so far, including glucose intolerance (**Fig 2A**; **S2A Fig**), cold intolerance (**Fig 2B**; **S2A Fig**), the tail tip defect (**Fig 2C**; **S2B Fig**), the excess SFA and MUFA/PUFA depletion in phosphatidylethanolamines (PEs; **Fig 2D and 2E**; **S2C Fig**) as well as the membrane rigidification phenotype measured using Fluorescence Recovery After Photobleaching (FRAP) when *paqr-2* mutants are cultivated on glucose or the SFA palmitic acid (**Fig 2H and 2I**). The *paqr-1(et52)* allele had no effect on the membrane fluidity of *paqr-2* mutants on normal plates or as a single mutant on normal plates or plates containing glucose (**S2E–S2H Fig**). These results suggest that *paqr-1(et52)* acts as a complete functional replacement for *paqr-2* and that it has no adverse effects under the conditions tested.

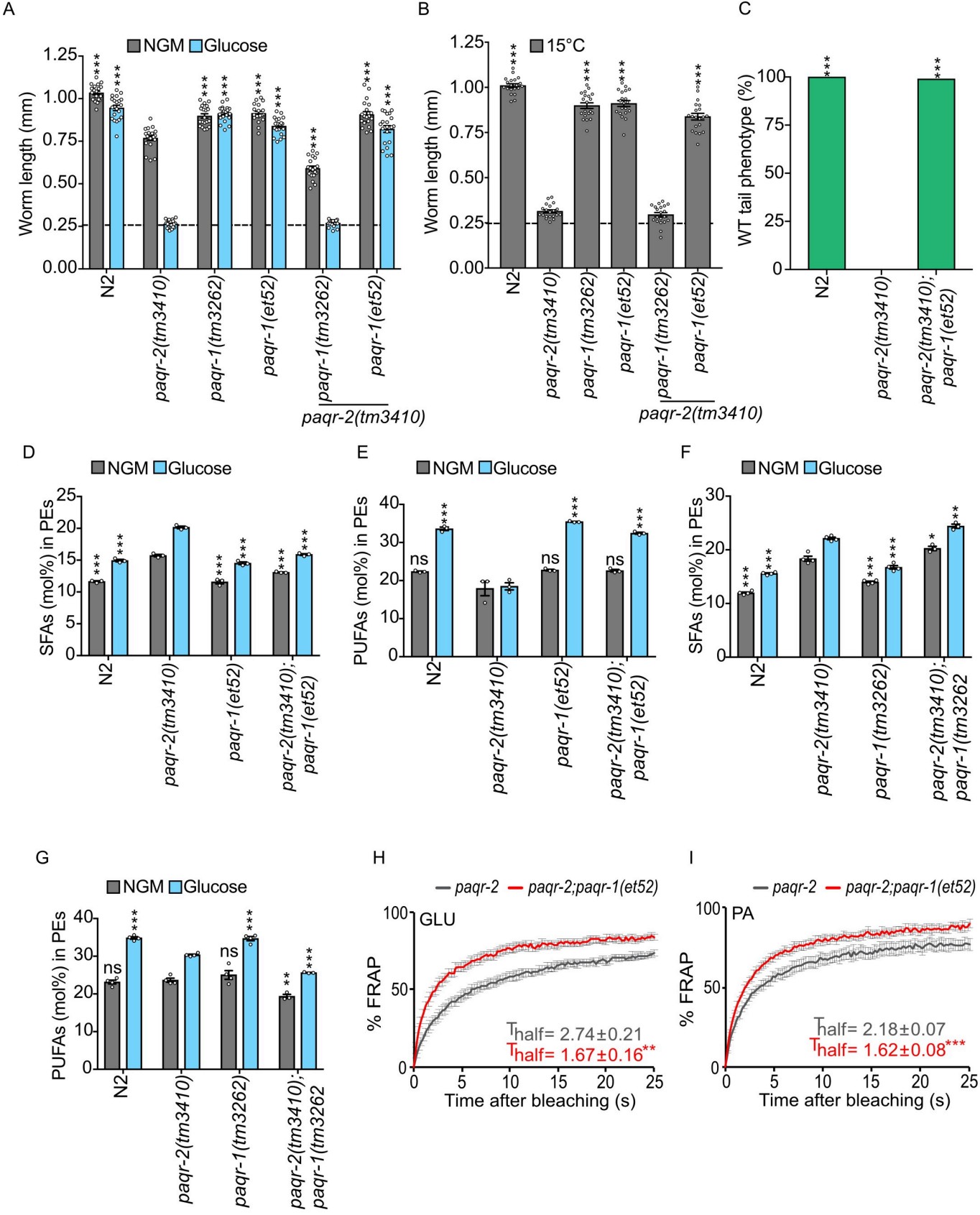

**Fig 2. The *paqr-1(et52)* allele suppresses several *paqr-2* mutant phenotypes. (A-B)** Length of worms placed as L1s on the indicated media/temperature and measured 72 or 144 hours later. Note that the *gof paqr-1(et52)* allele but not the null *paqr-1(tm3262)* allele suppresses the glucose and cold intolerance of the *paqr-2(tm3410)* null mutant. **(C)** Tail tip phenotype scored on 1-day old adults. Note that the *gof paqr-1(et52)* suppresses the tail tip defect of the *paqr-2(tm3410)* mutant; note that 100% of the *paqr-2* mutant worms show the withered tail tip phenotype and that this is completely suppressed by the *paqr-1(et52)* mutation. **(D-E)** The *gof paqr-1(et52)* allele suppresses the excess SFA and PUFA depletion in the PEs of the *paqr-2(tm3410)* null mutant; both the null *paqr-1(tm3262)* and *gof paqr-1(et52)* have normal levels of SFAs and PUFAs among PEs. **(F-G)** The double null *paqr-1(tm3262);paqr-2(tm3410)* mutant has a more severe excess SFA and PUFA depletion in PEs than the single null *paqr-2(tm3410)* mutant. **(H-I)** FRAP measurements showing that the *gof paqr-1(et52)* suppresses the low membrane fluidity found in *paqr-2(tm3410)* mutants grown on 20 mM glucose (GLU) or fed *E. coli* pre-loaded with 2 mM palmitic acid (PA). The dashed line in A indicates the approximate size of the L1 larvae at the start of the experiment. Significant differences compared to the *paqr-2* genotype are indicated where: * $p < 0.05$, ** $p < 0.01$ and *** $p < 0.001$ (ns: not significant).

In contrast, the loss-of-function allele *paqr-1(tm3262)* does not act as a *paqr-2* suppressor but rather slightly worsens several *paqr-2* mutant phenotypes such as poor growth on normal media (**Fig 2A**), phospholipid composition defects (**Fig 2F–2G**; **S2D Fig**), brood size (**S1F Fig**) and locomotion rate (**S1J Fig**). These results suggest that the wild-type *paqr-1* and *paqr-2* genes are partially redundant for these traits.

## *paqr-1(et52)* acts through the same pathway as *paqr-2*

Our previous work showed that IGLR-2 and PAQR-2 act as a fluidity sensor that signals through NHR-49 (a nuclear hormone receptor related to mammalian PPARα and HNF4β) and/or SBP-1 (the single worm homolog of the mammalian SREBPs) and MDT-15 (a mediator subunit that acts as a co-factor of NHR-49 and SBP-1) to promote the expression of Δ9 desaturases (summarized in **Fig 3A**) [18]. Using RNAi, we found that *mdt-15*, *sbp-1* and the *fat-5/-6/-7* desaturases (which share high sequence similarity and may all be silenced by siRNA against any one) are all required for the maximum activity of *paqr-1(et52)*, scored by measuring growth on plates containing 20 mM glucose (**Fig 3B**) or on normal media (**S3A Fig**). Since these are all *paqr-2* effectors [18], these results suggest that *paqr-1(et52)* is a version of *paqr-1* that acts via the same pathway as *paqr-2*. However, *paqr-2* and *nhr-49* must also have separate functions since the single loss-of-function mutants are viable but the double mutant is lethal (**Fig 3C and 3D**). Introducing the *paqr-1(et52)* into the *paqr-2;nhr-49* double mutant suppresses the lethality and tail tip defect (**Fig 3C and 3D**; **S3B Fig**) but does not suppress the glucose intolerance (glucose is here again used as an expedient way to provide an SFA-rich diet; **Fig 3C and 3D**) nor the cold intolerance (**S3C Fig**). These results suggest that *nhr-49* is an essential *paqr-2* and *paqr-1(et52)* downstream target for SFA tolerance but that it serves other functions as well. The *paqr-2;sbp-1* double mutant is lethal and we were unable to create a *paqr-2;sbp-1;paqr-1(et52)* triple mutant. This suggests that *paqr-1(et52)* cannot completely replace all *sbp-1*-independent functions of *paqr-2*.

## *paqr-2* inhibits *paqr-1(et52)* when *iglr-2* is absent

*paqr-2* is totally dependent on the presence of a functional *iglr-2* for its activity [17]. Similarly, *paqr-1(et52)* was unable to suppress the glucose intolerance of the *iglr-2(et34)* mutant (**Fig 3E and 3F**), though it suppressed the intolerance to cold (15°C), which is a milder stress (**Fig 3G**). Quite unexpectedly however, we found that *paqr-1(et52)* was able to suppress both glucose and cold intolerance in the *paqr-1(et52);paqr-2(tm3410);iglr-2(et34)* triple mutant (**Fig 3E–3G**), as well as its tail tip defect (**S3D Fig**). Taken together, these results suggest that the presence of *paqr-2* inhibits *paqr-1(et52)* when *iglr-2* is absent. Speculatively, it is possible that the PAQR-2 protein competes with PAQR-1(R109C) for a downstream factor only when IGLR-2 is absent, or that PAQR-2 can freely interact and interfere with PAQR-1(R109C) when IGLR-2 is absent. In either case, it is clear that the *iglr-2* gene is not required for the ability of *paqr-1(et52)* to rescue the *paqr-2* null mutant. Indeed, BiFC, a method that detects the interaction

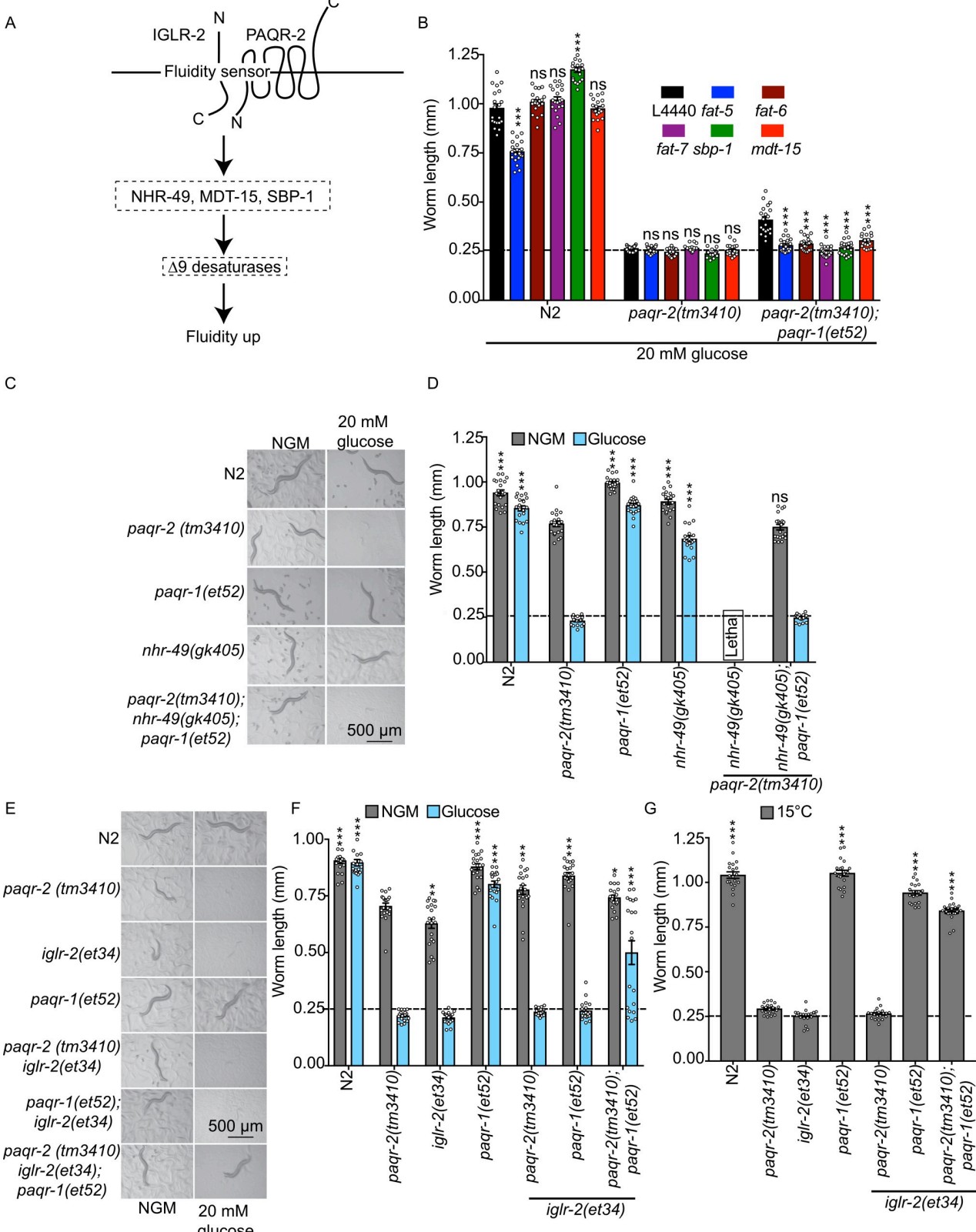

**Fig 3. The *gof paqr-1(et52)* allele acts via the same pathway as *paqr-2* but independently from *iglr-2*. (A)** Simplified PAQR-2/IGLR-2 membrane homeostasis pathway. **(B)** Length of worms placed as L1s on the indicated media/RNAi treatment and measured 72 hours later. Note that the ability of

*paqr-1(et52)* to suppress the poor growth defect of the null *paqr-2(tm3410)* mutant is dependent on the desaturases *fat-5, -6,* and *-7,* as well as the *sbp-1* and *mdt-15* transcription factors. Note also that these RNAi experiments are performed using the HT115 *E. coli* strain, which in itself is poorly tolerated by the *paqr-2(tm3410)* mutant[18]. **(C-D)** Images and length of worms placed as L1s on the indicated media and measured 72 hours later. Note that the *gof paqr-1(et52)* allele suppresses the lethality of the double null mutant *paqr-2(tm3410); nhr-49(gk405)* on normal growth media but not on plates containing 20 mM glucose. **(E-F)** Images and length of worms placed as L1s on the indicated media and measured 72 hours later. Note how the *gof paqr-1(et52)* allele suppresses the glucose intolerance of the double null mutant *paqr-2(tm3410) iglr-2(et34)* but not that of the single null mutant *iglr-2(et34)*. **(G)** Length of worms placed as L1s on NGM media and measured after 144 hours cultivation at 15˚C. Note how the *gof paqr-1 (et52)* allele suppresses the growth intolerance of both the double null mutant *paqr-2(tm3410) iglr-2(et34)* and the single null mutant *iglr-2(et34)*. The dashed lines in B, D, F and G indicate the approximate size of the L1 larvae at the start of the experiment. Significant differences compared to the *paqr-2* genotype are indicated where: $^{*}$ $p<0.05$, $^{**}p<0.01$ and $^{***}p<0.001$ (ns: not significant).

between PAQR-2 and IGLR-2 [17] (**S4A and S4B Fig**), failed to detect any interaction between IGLR-2 and either wild-type PAQR-1 or PAQR-1(R109C) (**S4C–S4E Fig**). This again is consistent with the PAQR-1 proteins acting independently of IGLR-2.

## Domain swapping suggests that the N-terminal cytoplasmic domain is regulatory

The *paqr-1(et52)* allele corresponds to a R109C amino acid substitution in the N-terminal cytoplasmic domain that is divergent between PAQR-1 and PAQR-2. Given the high degree of similarities in their transmembrane domains and the fact that the *paqr-1(et52)* allele affects an amino acid outside of the presumed enzymatic domain, it seems likely that PAQR-1 and PAQR-2 have similar enzymatic functions but are regulated differently via their cytoplasmic N-terminal domains. To test this more directly, we proceeded to swap various domains between the two proteins and added an HA tag at the N-terminus of the resulting chimeric proteins, which were all expressed from the *paqr-2* promoter to facilitate direct comparisons (**Fig 4A**). Two independent transgenic lines were generated for each construct, and their expression was confirmed using Western blots against the HA tag (**Fig 4B**). The constructs were then tested for their ability to rescue three phenotypes of *paqr-2* single and *paqr-2;iglr-2* double mutants: intolerance to cold, intolerance to glucose (which again is an expedient way to provide a SFA-rich diet[16]) and the tail tip morphology defect.

The full length PAQR-1(R109C) protein expressed from the *paqr-2* promoter, but not the full length wild-type PAQR-1, efficiently rescued growth of the *paqr-2* single mutant and *paqr-2;iglr-2* double mutant at 15˚C (**Fig 4C**) and on glucose (**S5A Fig**), as well as the tail tip morphology (**S5B Fig**). This confirms that addition of the HA tag at the N-terminus does not impair function and that driving expression of the PAQR-1(R109C) protein from the *paqr-2* promoter supports the ability of this *gof* allele to function effectively.

A chimeric protein composed of the N-terminal cytoplasmic domain of PAQR-1(R109C) and the transmembrane domains of PAQR-2, was able to rescue growth at 15˚C of the *paqr-2* single mutant, but not of the *paqr-2;iglr-2* double mutant (**Fig 4D**). This same chimeric protein did not rescue growth of the *paqr-2* single mutant on glucose nor its tail tip defect (**S5C and S5D Fig**). These results indicate that it is the transmembrane domains of PAQR-2 that dictate a requirement for the presence of the IGLR-2 protein, and that combining the N-terminal domain of PAQR-1(R109C) with the transmembrane domains of PAQR-2 results in a protein with reduced activity compared to the wild-type PAQR-2. Interestingly, it was not possible to create a *paqr-2;iglr-2* double mutant line expressing a chimeric protein consisting of the N-terminal cytoplasmic domain of wild-type PAQR-1 and the transmembrane domains from PAQR-2 (**Fig 4D**). This is consistent with the R109C mutation conferring increased activity to the N-terminal domain compared to that of the wild-type PAQR-1 protein.

Swapping the short extracellular C-terminal domain of the PAQR-2 protein with that of PAQR-1(R109C) resulted in a small reduction in the ability of the resulting chimeric protein

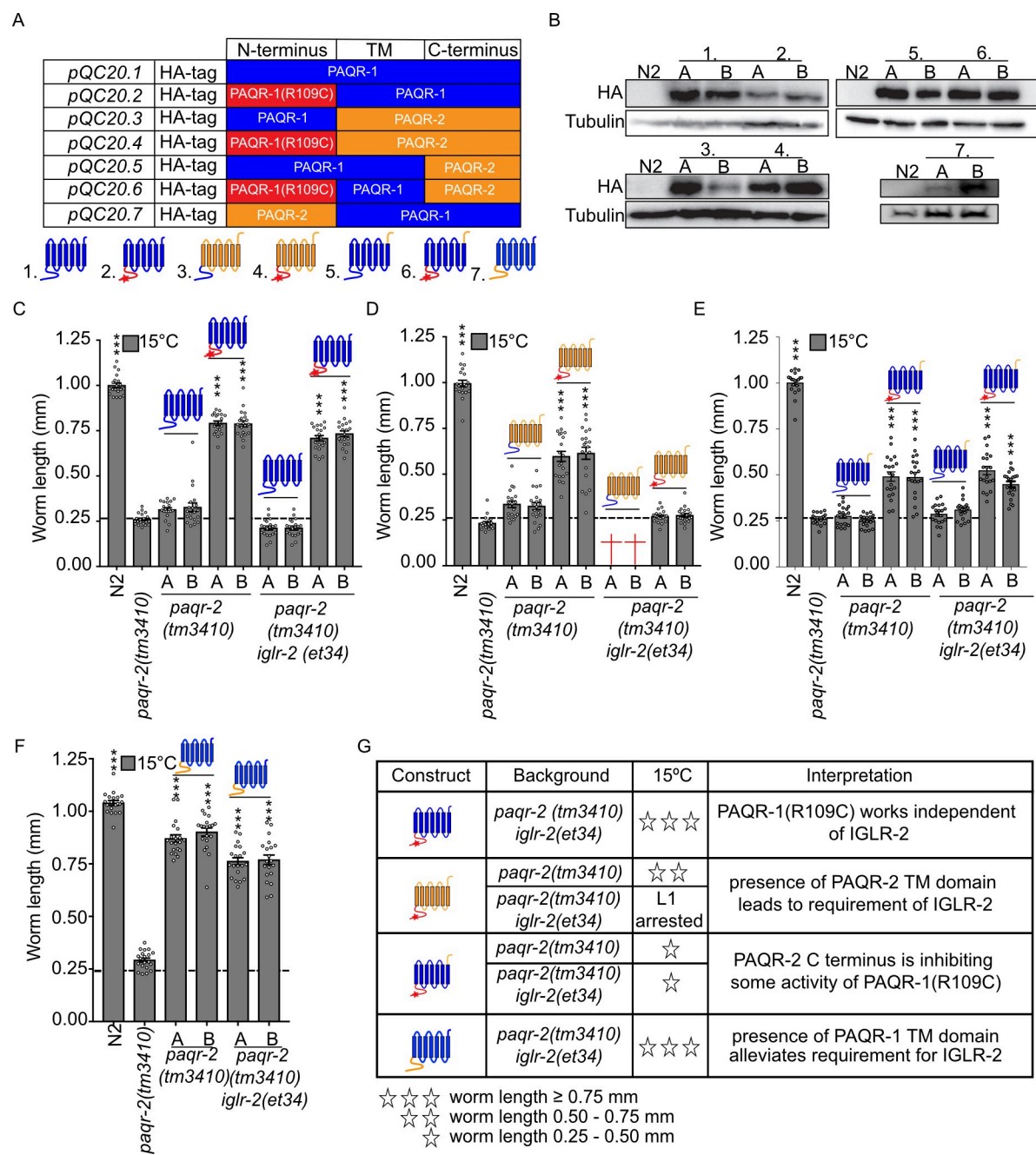

**Fig 4. Domain swapping experiments indicate that the intracellular domains of PAQR-2 and IGLR-2 are likely regulatory. (A)** Cartoon representation of the constructs used in this structure-function study, all driven by the *paqr-2* promoter. (**B**), Western blots showing expression of the tested constructs; two independent transgenic lines were generated for each construct, labelled A and B. (**C-F**) Length of worms placed as L1s on NGM media and measured after 144 hours cultivation at 15˚C. Note that *pQC20.2* is an excellent suppressor of the *paqr-2(tm3410) iglr-2(et34)* double null mutant (panel **C**) but that *pQC20.4* is not (panel **D**), which suggests that the transmembrane domains of PAQR-2 require IGLR-2 for activity. Crosses in **D** indicate likely lethality since it was impossible to obtain a strain with that genotype. (**G**) Main interpretations from the data in (**C-F**). The dashed lines in C-F indicate the approximate size of the L1 larvae at the start of the experiment. Significant differences compared to the *paqr-2* genotype are indicated where: * $p < 0.05$, ** $p < 0.01$ and *** $p < 0.001$ (ns: not significant).

to rescue the growth of *paqr-2* single mutants or *paqr-2;iglr-2* double mutants at 15˚C (compare **Fig 4E** with **Fig 4C**) or on glucose (compare **S5E Fig** with **S5A Fig**), as well as the tail tip phenotype (compare **S5F Fig** with **S5B Fig**). In other words, the activity of PAQR-1(R109C) is reduced when it carries the longer 44 amino acids C-terminal domain of PAQR-2 rather than its native 5 amino acids extracellular C-terminus.

Finally, a construct bearing the transmembrane domains of PAQR-1 fused to the cytoplasmic N-terminal domain of PAQR-2 was created and found to act as a potent *paqr-2* suppressor both in the presence or absence of IGLR-2 (**Fig 4F** and **S5G and S5H Fig**). This is an interesting result from which at least two conclusions may be drawn: 1) it is the transmembrane domains of PAQR-2 that impose a requirement for IGLR-2; and 2) fusing the PAQR-1 transmembrane domains to the cytoplasmic domain of PAQR-2 results in a protein that behaves as a PAQR-1 gain-of-function allele. A graphical summary of the main findings from the domain swapping experiments is presented in **Fig 4G**.

### AdipoR1 overexpression can compensates for AdipoR2 depletion

The structure-function studies in *C. elegans* suggest that the primary difference between PAQR-1 and PAQR-2 lies in their N-terminal regulatory cytoplasmic domains such that PAQR-1 can act without IGLR-2 while PAQR-2 requires an interaction with IGLR-2 for its activation. That PAQR-1 overexpression can suppress *paqr-2* mutant phenotypes suggests that it acts either alone or via an interaction partner that is not limiting for PAQR-1 activity. In either case, it is interesting to test whether this relationship is conserved between the human AdipoR1 and AdipoR2. For this purpose, we turned to HEK293 cells where the AdipoR1 and AdipoR2 proteins act as functional homologs of PAQR-2 and are required to prevent membrane rigidification when the cells are cultivated in the presence of palmitic acid [13–16, 35]. Here, we found that overexpression of either AdipoR1 or AdipoR2 in HEK293 cells is sufficient to prevent membrane rigidification in HEK293 cells challenged with 400 μM palmitic acid (**Fig 5A–5E**). Thus, either AdipoR1 and AdipoR2 act alone in mammalian cells, or the levels of their putative IGLR-2 orthologous partner is not limiting for their activity. Finally, we found that overexpression of AdipoR1 prevents membrane rigidification by 200 μM palmitic acid in cells where AdipoR2 has been silenced (**Fig 5F–5H**), thus echoing again the *C. elegans* findings where increased PAQR-1 activity compensates for loss of PAQR-2.

### Discussion

Our main findings are: 1) A single amino acid substitution in the cytoplasmic N-terminal domain of PAQR-1, i.e. R109C or R109A, is sufficient to act as a *gof* mutation. This suggests that the N-terminal cytoplasmic domain has a regulatory function. 2) The *paqr-1(et52) gof* allele acts through the same pathway as *paqr-2* and is a complete *paqr-2* mutant suppressor with no obvious adverse effect. Conversely, the *paqr-1(tm3262) lof* allele has no obvious phenotype on its own but worsens the *paqr-2* mutant phenotypes in double mutants. Taken together, these results suggest that *paqr-1* and *paqr-2* are partially redundant, but that *paqr-2* is more important. 3) The *paqr-1(et52) gof* allele suppresses the mutant phenotypes in *paqr-2; iglr-2;paqr-1(et52)* triple mutants better than it does in the *iglr-2;paqr-1(et52)* double mutant, and can also suppress the *paqr-2;iglr-2* mutant phenotypes when provided as a transgene. This suggests that the PAQR-1(R109C) protein does not require IGLR-2 for its activity but that PAQR-2 competes with PAQR-1(R109C) for a downstream factor only when IGLR-2 is absent, or that the PAQR-2 protein can interact and interfere with PAQR-1(R109C) when IGLR-2 is absent. 4) Domain swapping experiments suggest that the PAQR-2 transmembrane domains require the presence of IGLR-2 for their activity, but that some essential PAQR-2

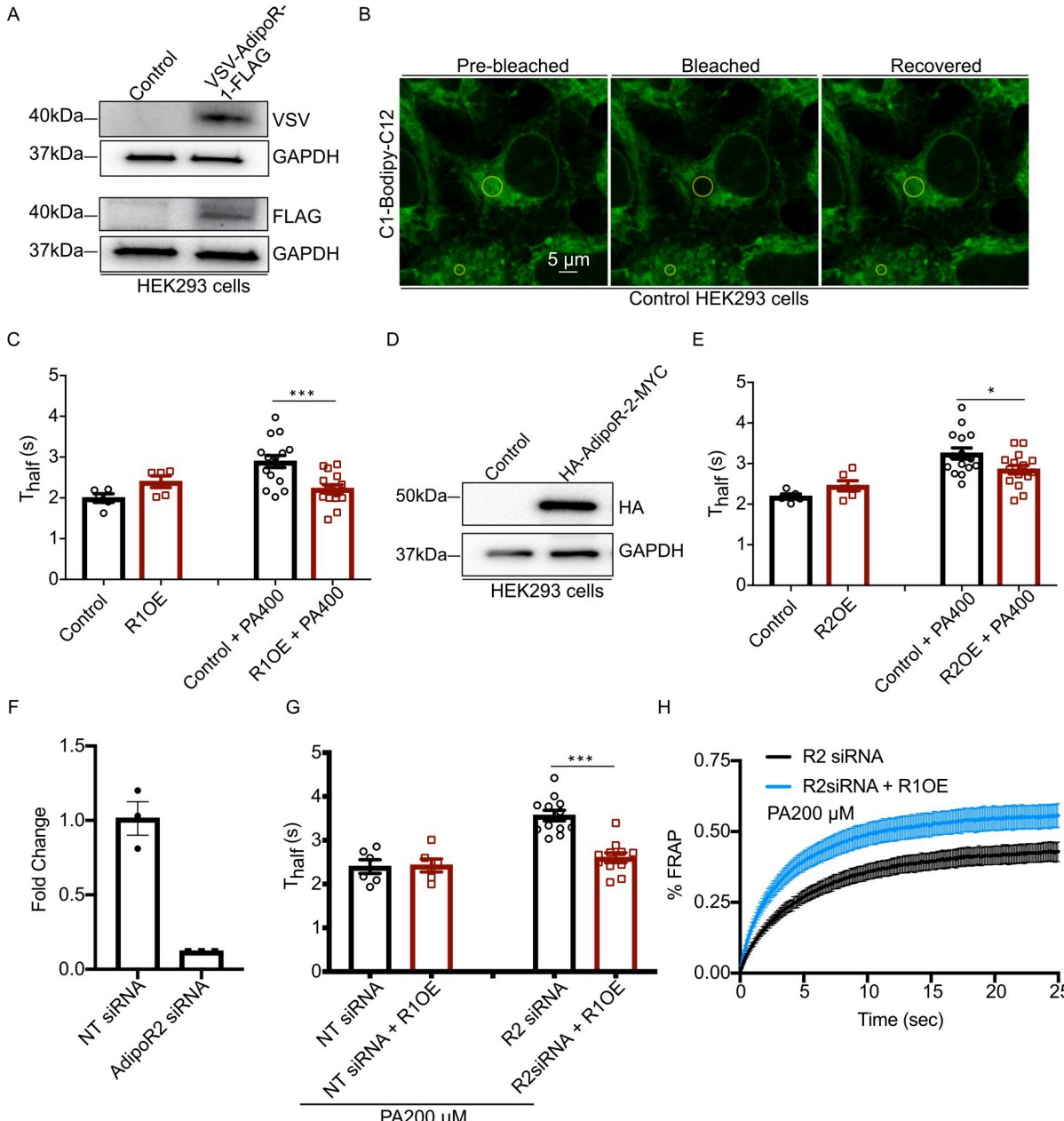

**Fig 5. Overexpression of AdipoR1 or AdipoR2 compensates for the absence of AdipoR2 in mammalian cells. (A)** Western blot detection of VSV-AdipoR1-FLAG transiently expressed in HEK293 cells, with GAPDH used as a loading control. **(B)** Snapshots from a typical FRAP experiment, with the bleached areas (large circle) and reference area (small circle) indicated before and after bleaching. **(C)** $T_{half}$ from FRAP experiments showing that the addition of 400 μM palmitic acid causes membrane rigidification and that this effect is attenuated by AdipoR1 overexpression (R1OE). **(D)** Western blot detection of HA-AdipoR2-MYC transiently expressed in HEK293 cells, with GAPDH used as a loading control. **(E)** $T_{half}$ from FRAP experiments showing that the addition of 400 μM palmitic acid causes membrane rigidification and that this effect is attenuated by AdipoR2 overexpression (R2OE). **(F)** Efficiency of the AdipoR2 siRNA knockdown normalized to non-target siRNA (NT siRNA) and quantified using QPCR. **(G)** $T_{half}$ from FRAP experiments showing that the addition of overexpression of AdipoR1 (R1OE) prevents membrane rigidification in cells where AdipoR2 has been silenced by siRNA. **(H)** FRAP curves for the AdipoR2 siRNA-treated samples from G; note the improved fluorescence recovery rate when AdipoR1 is overexpressed (R1OE). Significant differences are indicated where: * $p < 0.05$ and *** $p < 0.001$.

activity is still present in the absence of IGLR-2 if the PAQR-1(R109C) N-terminal cytoplasmic domain is attached to the PAQR-2 transmembrane domains. This suggests that the interaction of IGLR-2 and PAQR-2 via their transmembrane domains may allow IGLR-2 to activate PAQR-2 via its regulatory N-terminal cytoplasmic domain. And 5) Overexpression of either AdipoR1 or AdipoR2 confers increased protection against palmitic acid-induced membrane rigidification in HEK293 cells.

The present work sheds new light on the roles of major protein domains in PAQR-1 and PAQR-2. In particular, it is clear that PAQR-1(R109C) and PAQR-2 act through the same pathway and can carry out very similar roles for the maintenance of membrane homeostasis, which suggests that the key physiological functions of these proteins reside within their highly conserved transmembrane domains and/or the conserved membrane-proximal 80 amino acids. The large cytoplasmic N-terminal domain, which is highly divergent between the two proteins, likely has a regulatory function, for example by blocking access to the cytoplasm-facing cavity where the presumed hydrolytic site is located. A similar suggestion was made for the cytoplasmic domain of the AdipoRs: the published crystal structures of the AdipoRs does not include the full-length N-terminal domain but the small portion (i.e. residues 89–120 for AdipoR1) that was included seem to obstruct the access to the cytoplasm-facing cavity, which prompted the authors to suggest that "a much larger opening at helices III–VII would be uncovered on the cytoplasmic side, if the NTR (N-terminal region) was displaced from its present position" [21]. Conceptually, this type of steric regulation is similar to the "ball and chain" mechanism first proposed to regulate sodium channels in neurons, where the cytoplasmic domain of the channel acts as a ball attached by a flexible motif to the channel, which it can block [36–38]. It will be interesting in the future to test whether novel *gof* alleles may be created by expressing PAQR-1 or PAQR-2 with truncations in their cytoplasmic N-terminal domains, which incidentally harbors membrane localization motifs in the mammalian AdipoRs [39, 40].

PAQR-1, which does not require IGLR-2, may have a low level of constitutive activity; this would explain the observation that over-expression of the wild-type PAQR-1 partially suppresses *paqr-2* mutant phenotypes. The PAQR-1(R109C) or PAQR-1(R109A) mutations may displace the regulatory N-terminal domain and thus cause a larger/more frequent access to the active site, and thus act as *gof* mutations. Arginine residues are important mediators of protein-protein interactions in a wide range of biological processes thanks to the positively charged guanidinium group (-C- (NH2)2$^+$) with five hydrogen bond donors [41–44]. Positively charged residues such as arginine are preferentially localized on the surface of proteins where they facilitate the interaction with negatively charged surfaces: about 40% of the salt-bridges in proteins involve ion pairs between arginine and the carboxylate groups of acidic amino acids [45], and the positive charge is also important for planar stacking and polar interactions with aromatic residues [46, 47]. Unsurprisingly, the number of arginines at protein interfaces is larger than expected from random distribution [48], and guanidinium groups are also significantly overrepresented at protein-protein interfaces, as opposed to other polar and charged groups [49]. It is therefore likely that the R109C and R109A substitutions in PAQR-1 have important structural consequences, for example disrupting intramolecular or intermolecular interactions important for the proposed "ball and chain" mechanism. Such a mechanism is also consistent with the *gof* allele created by fusing the transmembrane domains of PAQR-1 to the cytoplasmic domain of PAQR-2, presumed here to poorly inhibit access to the catalytic site. Our finding that overexpression of AdipoR1 or AdipoR2 is sufficient to protect HEK293 cells from palmitic acid-induced rigidification echoes those of Kupchak *et al.* who showed that AdipoR1 expression in yeast support its activity and mimic that of the yeast homologs [50], and of Holland *et al.* who showed that overexpression of AdipoR1 and

AdipoR2 in liver also leads to their increased activity, as measured by the ceramidase activity in extracts as well as improved glucose and lipid homeostasis [29]. Clearly, the *C. elegans* PAQR-1 and the human AdipoR proteins can act in a dose-dependent manner. This suggests that PAQR-1 and the AdipoRs have intrinsic basal activity that is not strictly regulated by the membrane environment. This is interesting given that the tissue expression levels of AdipoR1 and AdipoR2 are quite variable. For example, and although all tissues appear to express either one or both of the AdipoRs, the retina is clearly a tissue with exceptionally high levels of AdipoR1 expression [51]. Not surprisingly then, the retina shows a severe depletion of membrane UFAs associated with retinitis pigmentosa both in mouse models and in human patients [51–53].

In contrast to PAQR-1, the activity of PAQR-2 is dependent on the presence of IGLR-2 and is only required under conditions of membrane rigidification, such as low temperature or an SFA-rich diet. The domain swapping experiments described here suggest that IGLR-2 and PAQR-2 interact primarily via their transmembrane domains but that an important consequence of that interaction occurs at the level of their cytoplasmic domains. Speculatively, docking of IGLR-2 onto PAQR-2 via the transmembrane domains could allow the cytoplasmic domain of IGLR-2 to cause a conformational change within the cytoplasmic domain of PAQR-2 that results in its activation, i.e. displacement of the "ball" that regulates access of substrates (e.g. ceramides as per [22, 28, 29]) to the active site (see model in **Fig 6**). IGLR-2 is related to the mammalian LRIG protein family, which are characterized by the presence of a single transmembrane domain, one or more immunoglobulin domains and leucine-rich repeats in their large extracellular N-terminal domain, and a cytoplasmic domain that varies in size and sequence. Many LRIG proteins act as regulators of signaling transmembrane proteins. The LRIG1, -2 and -3 proteins for example are important regulators of receptor tyrosine

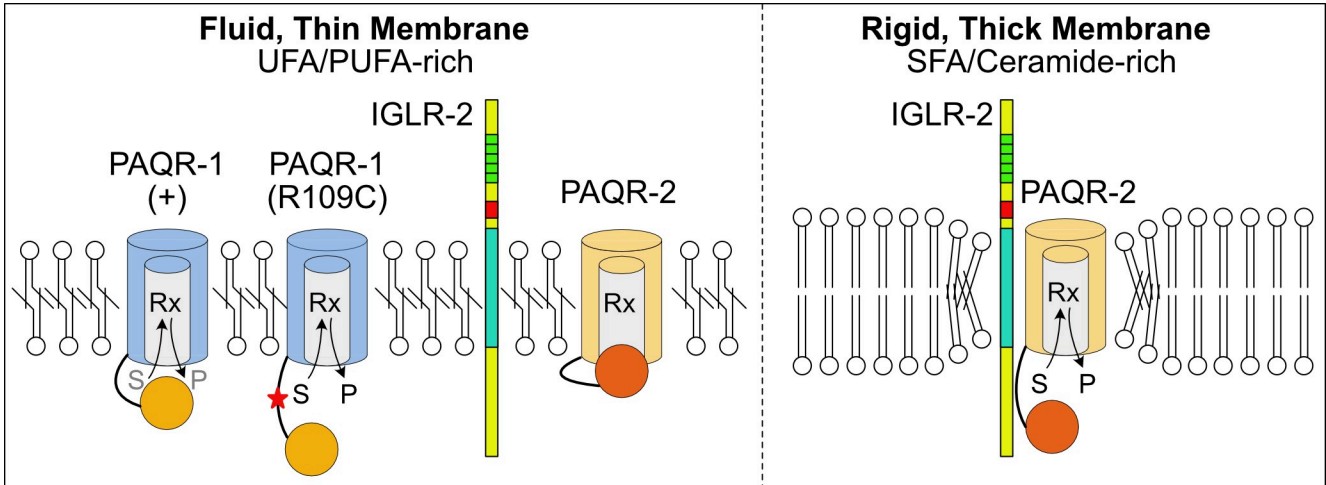

**Fig 6. Speculative mechanism of PAQR-1/PAQR-2 regulation.** We propose that the cytoplasmic domain (orange/red circles) can regulate access to the catalytic site (Rx) located within the cavity formed by the seven transmembrane domains of PAQR-1/2. In PAQR-1(+), the regulatory domain partially blocks access, resulting in a small rate of conversion of the substrate (S) to signaling product (P). The R109C mutation causes a conformational change in the cytoplasmic regulatory domain, providing greater access to the catalytic site, which results in more substrates being converted to signaling products, which explains the gain-of-function nature of this mutation. The activity of PAQR-1 is not affected by membrane composition, but is rather dose-dependent. In fluid/thin membranes, the PAQR-2 and IGLR-2 proteins readily diffuse within membranes and do not form stable complexes, though they may occasionally interact with each other; the PAQR-2 cytoplasmic inhibitory domain blocks access to the active site under these conditions. In rigid/thick membranes, the PAQR-2 protein causes local deformations of the lipid bilayer that stabilize its interaction with IGLR-2, forming energetically favored signaling nexus; the PAQR-2 cytoplasmic inhibitory domain is displaced through its interaction with IGLR-2, freeing access to the catalytic site. The model attempts to explain the regulation of the hydrolase activity (e.g. ceramidase) that likely provides a ligand for downstream targets; PAQR-1 and PAQR-2 may have additional common interaction partners (not depicted) that explain why PAQR-2 can act as an inhibitor of PAQR-1(R109C) when IGLR-2 is absent.

kinases [54–56]. The mammalian LRIG protein AMIGO, which structurally is the most similar to *C. elegans* IGLR-2, regulates Kv2.1 voltage-gated potassium channels by facilitating their clustering [57]. IGLR-2 could also act as a promoter of PAQR-2 multimerization to create a signaling nexus. Membrane thickening, which typically accompanies rigidification, could promote PAQR-2/IGLR-2 clustering because of the energy costs of membrane deformation around the fixed-length transmembrane domains of the proteins in a situation analogous to that found in GpA helix-dimers [58, 59] or Ire1[12, 60], another membrane fluidity sensor. Considerations regarding clustering are interesting because several independent lines of evidence suggest that the mammalian AdipoR1 and AdipoR2 proteins can form homo- and heteromultimers; these include co-immunoprecipitation of tagged AdipoR1 and AdipoR2 [61], bifluorescence complementation of AdipoR1 in HEK293 cells [62] and fluorescence resonance energy transfer (FRET) also in HEK293 cells [63]. It will be interesting in the future to determine whether a mammalian homolog of IGLR-2 facilitates the multimerization of the AdipoRs. A better understanding of this membrane homeostasis pathway is important because of its potential therapeutic value in many disease contexts where abnormal membrane properties have been noted, including diabetes [64, 65] and cancer [66–69].

## Materials and methods

### *C. elegans* strains and cultivation

The wild-type *C. elegans* reference strain N2 and the mutant alleles studied (except for the novel *paqr-1(et52)*) created in the present study) are available from the *C. elegans* Genetics Center (CGC; MN; USA). Unless otherwise stated, the *C. elegans* strains maintenance and experiments were performed at 20˚C using the *E. coli* strain OP50 as food source, which was maintained on LB plates kept at 4˚C (re-streaked every 6–8 weeks) from which single colonies were picked for overnight cultivation at 37˚C in LB medium, then used for seeding NGM plates [70]. OP50 stocks were kept frozen at -80˚C and new LB plates were streaked every 3–4 months. NGM plates containing 20 mM glucose were prepared using a 1 M stock solution that was filter sterilized then added to the molten NGM media.

The *paqr-1(syb1401)* allele in which a HA tag is fused in frame to the start codon of *paqr-1* was created by Suny Biotech Co (Fuzhou City, China) using CRISPR/Cas9. The altered sequence is as follows (HA coding sequence is in upper case; underlined sequence is synonymous mutation and endogenous sequence of *paqr-1* is in lower case):

catgatcactacaatgtattctatcatctttctttcattttttgcaaacatgaagacaaagttcatatttcaggcaaagaatgTACC CATATGATGTCCCGGATTACGCTaatccagatgaggtcaaccgagcccttgggcactacctcaatgacgctgatt caggcgaattggttgtcgaggacagcacaactgtacaggtaggattgaagaagaaaaataatattgatgttaaaattaaaaaacgttca tatttttctaaattatatatcaatt.

The *paqr-1(syb364)* allele in which a HA tag is fused in frame to the start codon of *paqr-1 (et52)* was created by Suny Biotech Co (Fuzhou City, China) using CRISPR/Cas9. The altered sequence is as follows (HA coding sequence is in upper case; underlined sequence are synonymous mutations and the endogenous sequence of *paqr-1(et52)* is in lower case):

catgatcactacaatgtattctatcatctttctttcattttttgcaaacatgaagacaaagttcatatttcaggcaaagaatgTACC CATATGATGTCCCGGATTACGCTaatccagatgaggtcaaccgagcccttgggcactacctcaatgacgctgatt-caggcgaattggttgtcgaggacagcacaactgtacaggtaggattgaagaagaaaaataatattgatgttaaaattaaaaaacgttca tatttttctaaattatatatcaatt.

The *paqr-2(syb1401)* allele in which a HA tag is fused in frame to the start codon of *paqr-2* was created by Suny Biotech Co (Fuzhou City, China) using CRISPR/Cas9. The altered sequence is as follows (HA coding sequence is in upper case; and the endogenous sequence of *paqr-2* is in lower case):

atccgctttattctctcacagttccgattttatttgatttttttctggaatttcttatattctcggttgaaaaaaatttaaaaactaaaattcagctt
taacaaaatgTACCCATATGATGTCCCGGATTACGCTgaggaagatgacgtggaatcggcaacaccggcgg
aatcgcaaaaactttttgcaaaaaagcgttcgaaattcgtttgacgag.

## Screen for suppressors of SFA intolerance and whole genome sequencing

*paqr-2(tm3410) mdt15(et14)* double mutant worms were mutagenized for 4 hours by incubation in the presence of 0.05 M ethyl methane sulfonate according to the standard protocol [70]. The worms were then washed and placed on NGM plate. Two hours later, vigorous hermaphrodite L4 animals were transferred to new NGM plates. Five days later, F1 progeny were bleached, washed and their eggs were allowed to hatch overnight in M9 (22 mM KH2PO4, 42 mM Na2HPO4, 85.5 mM NaCl and 1 mM MgSO4). The resulting L1 larvae were transferred to new plates containing 20 mM glucose and then screened 72 hours later for fertile adults, which were picked onto new NGM plates for further analysis.

The isolated suppressor alleles were outcrossed 4 to 6 times prior to whole genome sequencing and 10 times prior to their phenotypic characterization or use in the experiments presented in the manuscript.

The genomes of suppressor mutants were sequenced to a depth of 25-40x as previously described [71]. Differences between the reference N2 genome and that of the mutants were sorted by criteria such as non-coding substitutions, termination mutations, splice-site mutations etc. [72] For each suppressor mutant, one or two hot spots, that is small genomic area containing several homozygous mutations, were identified, which is in accordance to previous reports [73]. Mutations in the hot spot that were still retained after 10 outcrosses were considered candidate suppressors and tested experimentally as described in the text.

## Pre-loading of *E. coli* with palmitic acid

A stock of 0.1 M palmitic acid (Sigma) was dissolved in ethanol and diluted in LB media to a final concentration of 2 mM, inoculated with OP50 bacteria, then shaken overnight at 37˚C. The bacteria were then washed twice with M9 to remove traces of palmitic acid and growth media, diluted to equal concentrations based on optical density ($OD_{600}$), concentrated 10x by centrifugation, dissolved in M9 and seeded onto NGM plates lacking peptone (200 μl/plate). Worms were added the following day.

## Growth, tail tip scoring and other *C. elegans* assays

For length measurement studies, synchronized L1s were plated onto test plates seeded with *E. coli*, and worms were mounted and photographed 72, 96 or 144 hours later. The length of 20 worms were measured using ImageJ [74]. Quantification of the withered tail tip phenotype was done on synchronous 1-day old adult populations, that is 72 h post L1 (n >100) [18]. Other assays starting with 1 day old adults have also previously been described in details: total brood size (n = 5) [30], lifespan (n = 100)[30], defecation (n = 5; average interval between ten defecation was determined for each worm) [75], pharyngeal pumping rate (n = 25, each monitored for 20s)[76], speed of locomotion (n = 5) [77].

## RNAi in *C. elegans*

All strains were grown on control L4440 RNAi bacteria for one generation at 20˚C, then synchronized and L1s placed onto assay RNAi, incubated at 15˚C and scored on day 6. Feeding RNAi clones were from the Ahringer RNAi library and were sequenced to confirm their identity, and used as previously described [78].

## Fluorescence recovery after photobleaching (FRAP) in *C. elegans* and HEK293 cells

FRAP experiments in *C. elegans* were carried out using a membrane-associated prenylated GFP reported expressed on intestinal cells as described previously and using a Zeiss LSM700inv laser scanning confocal microscope with a 40X water immersion objective [16, 17]. Briefly, the GFP positive membranes were photobleached over a circular area (seven-pixel radius) using 20 iterations of the 488 nm laser with 50% laser power transmission. Images were collected at a 12-bit intensity resolution over 256 x 256 pixels (digital zoom 4X) using a pixel dwell time of 1.58 μsec, and were all acquired under identical settings. For FRAP in mammalian cells, HEK293 cells were stained with BODIPY 500/510 C1, C12 (Invitrogen) at 2 μg/ml in PBS for 10 min at 37˚C [16]. FRAP images were acquired with an LSM880 confocal microscope equipped with a live cell chamber (set at 37˚ and 5% $CO_2$) with a 40× water-immersion objective as previously described [16]. The recovery of fluorescence was traced for 25 s. Fluorescence recovery and $T_{half}$ were calculated as described previously [17].

## Western blot for *C. elegans*

Crowded plates of various *C. elegans* strains were harvested and washed twice in M9 and then lysed using the following lysis buffer: 25 mM Tris (pH 7.5), 300 mM NaCl, 1% triton X-100 and 1X protease inhibitor. 25 μg of total protein was loaded in each lane. For transgenic animals, 25 worms expressing GFP (used as a transformation marker) were picked, boiled in sample buffer (4x Laemmli sample buffer, BIO-RAD) and loaded on an SDS gel. For detection of protein the nitrocellulose membranes (GE Health care) were blocked with 5% skimmed milk (Blotting-Grade Blocker, BIO-RAD) diluted in PBS-T. Antibody dilutions (primary antibody: rabbit monoclonal anti-HA antibody (C29F4; Cell Signaling Technology) 1:5000, mouse monoclonal anti-TUBULIN (T5168; SIGMA) and secondary antibodies: swine anti-rabbit HRP (1:3000, Dako) or goat anti-mouse HRP (1:3000, Dako)) was done in 5% skimmed milk in TBST and washes were carried out in PBS-T. Detection of the bound antibodies was performed using an ECL detection kit (Immobilon Western; Millipore) and visualized with a digital camera (VersaDoc; Bio-Rad).

## Construction of plasmids

**pQC20.10 and pQC20.11: *paqr-1* and *paqr-1(et52)* rescue constructs.** *paqr-1* and *paqr-1 (et52)* rescue constructs were created using the following primers 5´ATCCAATTTGCCCCAC TGAAT 3´´and 5´ CACAAAACTCTAGACTACTGG 3´ to amplify the *paqr-1* and *paqr-1 (et52)* alleles and 2 kb of upstream regulatory sequence using genomic DNA from lysed worms as template. The resulting 5 kb PCR products were cloned into the pCR XL-TOPO vector using TOPO XL PCR cloning kit (Invitrogen). The resulting plasmids, *pQC20.10* and *pQC20.11*, were injected into N2 worms at 10 ng/μl together with 3 ng/ μl *pPD118.33* (*Pmyo-2::GFP*) used as a transformation marker, a gift from Andrew Fire (Addgene plasmid # 1596) [79].

**pQC20.8 and pQC20.9: *paqr-1* and *paqr-1(et52)* translational reporters.** The translational reporters were generated with a Gibson assembly cloning kit (NEB) with the following two fragments: (1) The promoter and the coding sequence was amplified from the rescue construct *pQC20.10* and *pQC20.11* with the following primers 5´GCATGGATGAACTATACAA AAATCCAGATGAGGTCAATCG-3´and 5´- CTCCTTTACTCATTGATGCCATTCTTTGC CTGAAATATGAAC-3´, (2) GFP was amplified from the plasmid *Ppaqr-2:paqr-2:GFP[30]* using the primers 5´- GTTCATATTTCAGGCAAAGAATGGCATCAATGAGTAAAGGAG-3´ and

5´- CGATTGACCTCATCTGGATTTTTGTATAGTTCATCCATGC-3´. The resulting plasmids, *pQC20.8* and *pQC20.9* support expression of *Ppaqr-1::PAQR-1WT::GFP* and *Ppaqr-1::PAQR-1 (et52)::GFP*, respectively, and were injected into N2 worms at 10 ng/μl together with the plasmid 30 ng/μl *PRF4*, which carries the dominant *rol-6(su1006)* transformation marker [80].

## Domain swapping constructs

**pQC20.1 and pQC20.2.**   Constructs carrying full length *paqr-1* or *paqr-1(et52)* driven from the *paqr-2* promoter were generated with a Gibson assembly cloning kit (NEB) by assembly of the following fragments: (1) The promoter and the backbone was amplified from *Ppaqr-2:paqr-2:GFP* plasmid[30] using the primers 5´-CGTCTGAACGAACAATGTCCAGTTAGATAGGGTAGATTTGTTTC-3´and 5´-GGATTAGCGTAATCCGGGACATCATATGGGTACATTTTGTTAAAGCTGAA-3´and (2) the coding sequences were amplified from the rescue constructs using the following primers 5´-TTCAGCTTTAACAAAATGTACCCATATGATGTCCCGGATTACGCTAATCC-3´ and 5´- GAAACAAATCTACCCTATCTAACTGGACATTGTTCGTTCAGACG-3´. The resulting *pQC20.1* and *pQC20.2* plasmids were injected into N2 worms at 10 ng/μl together with 40 ng/μl *sur-5::gfp [81]* used as a transformation marker.

**pQC20.3 and pQC20.4.**   The cytoplasmic N terminal domain of *paqr-1* and *paqr-1(et52)* were fused to the transmembrane and C-terminal domains of *paqr-2* using a Gibson assembly cloning kit (NEB) with the following fragments: (1) the *paqr-2* promoter and coding sequence for the PAQR-1 N-terminal domain was amplified from the *pQC20.1* and *pQC20.2* plasmids using the primers 5´-CTAAAATTCAGCTTTAACAAAATGAATCCAGATGAGGTCAATCGAGCCCTTGG-3´and 5´-CATATGAGTCCAAATGTTTCCTGTTTCCGTGTGCAGTGACCAAATA-3´and (2) the *paqr-2* transmembrane and C-terminal coding sequences were amplified from the *Ppaqr-2:paqr-2:GFP* plasmid[30] using the primers 5´- TATTTGGTCACTGCACACGGAAACAGGAAACATTTGGACTCATATG-3´ and 5´- CCAAGGGCTCGATTGACCTCATCTGGATTCATTTTGTTAAAGCTGAATTTTTAG-3´. The resulting *pQC20.3* and *pQC20.4* plasmids were injected into N2 worms at 10 ng/μl together with 40 ng/μl *sur-5::gfp[81]* used as a transformation marker.

**pQC20.5 and pQC20.6.**   The N-terminal cytoplasmic domain and the transmembrane domains encoded by the *paqr-1* wild-type and *et52* alleles were fused to the C-terminus of *paqr-2* using a Gibson assembly cloning kit (NEB) with the following fragments: (1) the promoter and N-terminal/transmembrane domain-coding sequences were amplified from the *pQC20.1* and *pQC20.2* constructs using the primers 5´-TTCAGTTGATCCGGGCAGGATCCTTTGTTCAGACGAGCAAATGC-3´ and 5´-ACCGGCGGATGTTGGTTTATGAGGTAGATTTGTTTCCAAT-3´ and (2) The *paqr-2* -C terminus coding sequence was amplified from the *pQC20.3* and *pQC20.4* plasmids using the following primers 5´-GCATTTGCTCGTCTGAACAAAGGATCCTGCCCGGATCAACTGAA-3´ and 5´-ATTGGAAACAAATCTACCTCATAAACCAACATCCGCCGGT-3´. The resulting *pQC20.5* and *pQC20.6* plasmids were injected into N2 worms at 10 ng/μl together with 40 ng/μl *sur-5::gfp[81]* used as a transformation marker.

**pQC20.7.**   The N terminal cytoplasmic domain encoded by *paqr-2* was fused to transmembrane domains and C terminus of paqr-1 wild-type using a Gibson assembly cloning kit (NEB) with the following fragments: (1) the N terminal of *paqr-2* was amplified from the wild-type worms that are CRISPR modified and carrying an HA tag at the N terminus (soon after ATG) using the following primers 5'CTAAAATTCAGCTTTACAAAATGTACCCATATGATGTCCCGGATTACGCT 3'and 5'GTGTCCAAATGTTACCAGTTTCAGTGTGCAATGCAAAAATA 3'and (2) the paqr-1 transmembrane domains and C terminus was amplified from

*pQC20.1* plasmid using the following primers 5'AGCGTAATCCGGGACATCATATGGGTA CATTTTGTTAAAGCTGAATTTTAG 3'and 5'TATTTTTGCATTGCACACTGAAACTGG TAACATTTGGACAC 3'. The resulting *pQC20.7* plasmid was injected into N2 worms at 10 ng/μl together with 40 ng/μl *sur-5::gfp* used as a transformation marker.

### *pQC20.12*, a PAQR-1(R109A) expression construct

Arginine at position 109 of the PAQR-1 coding sequence was replaced by alanine using a site directed mutagenesis kit (NEB). The primers used were 5′-GCCCGTAAAAAGGGAGGG CAAT-3′and 5′-ATATCTGAAAAATTATGCG-3′ and the wild-type *paqr-1* rescue construct (*pQC20.10*) was used as template. The resulting plasmid, *pQC20.12*, was injected into N2 worms at 10 ng/μl together with 40 ng/μl *sur-5::gfp[81]* used as a transformation marker.

### Bimolecular fluorescence complementation (BiFC) constructs and analysis

*pCE-IGLR2-VC155*, *pCE-VN173-PAQR2* and *pCE-VN173-PAQR1 WT* plasmids were used from a previously published paper [17]. *pCE-VN173-PAQR1 R109C* construct was generated by Q5 site directed mutagenesis kit (NEB) using the following primers 5'TTTCAGATATTGTCGTAAAAA 3'and 5'AATTATGCGAATTTTTAAAA 3'. Wild type version of PAQR-1 plasmid was used as a template. The different combinations of BiFC plasmids were injected into N2 worms at 15 ng/μl each, together with *pRF4(rol-6)* at 100 ng/μl as previously described [82]. Expression of the BiFC constructs were induced by heat shocks of 2.5 h and 1.5 h at 33˚C, with 2 h recovery at 20˚C in between. Scoring of fluorescence was preformed after 16 h of recovery at 20˚C.

### *C. elegans* lipidomics

For worm lipidomics, samples were composed of synchronized L4 larvae (one 9 cm diameter plate/sample; each treatment/genotype was prepared in four independently grown replicates) grown overnight on OP50-seeded NGM or NGM containing 20 mM glucose. Worms were washed 3 times with M9, pelleted and stored at -80˚C until analysis. For lipid extraction, the pellet was sonicated for 10 minutes in methanol and then extracted according to BUME method [83]. Internal standards were added in the chloroform phase during the extraction. Lipid extracts were evaporated and reconstituted in chloroform: methanol (1:2) with 5 mM ammonium acetate. This solution was infused directly (shotgun approach) into a QTRAP 5500 mass spectrophotometer (ABSciex) equipped with the Nanomate Triversa (Advion Bioscience) as described previously[84]. Phospholipids were measured using multiple precursor ion scanning [85, 86]. The data was evaluated using the LipidProfiler software [85]. The full lipidomics data is provided as in the file **S1 Lipidomics**.

### Cultivation of HEK293 cells

HEK293 (identity verified by STR profiling and tested free of mycoplasma) cells were grown in DMEM containing 1 g/l glucose, pyruvate, and GlutaMAX and supplemented with 10% FBS, 1% nonessential amino acids, 10 mM HEPES, and 1% penicillin and streptomycin (all from Life Technologies) at 37˚C in a water-humidified 5% CO2 incubator. Cells were sub-cultured twice a week at 90% confluence. For FRAP, the cells were seeded in glass-bottom dishes (Ibidi) pre-coated with 0.1% porcine gelatin (Sigma-Aldrich).

   **AdipoR1/2 constructs and over-expression.** The human AdipoR1 cDNA (NP_001277482.1) was cloned between the NheI and BspEI sites of the pIRESneo2 vector (Clontech, Takara Bio) including the N-terminal VSV-tag and C-terminal FLAG-tag. The human AdipoR2 cDNA (NP_001362293.1) was cloned between the Nhe I and XmaI sites of

the pIREShyg2 vector (Clontech, Takara Bio) including the N-terminal HA tag and C-terminal c-Myc tag. HEK293 cells were transfected using Viromer Red according to the manufacturer's instructions 1X protocol (Lipocalyx).

## siRNA in HEK293 cells

The predesigned siRNA: AdipoR2 J-007801-10 and Non-target D-001810-10 were purchased from Dharmacon. HEK293 cell transfection was performed in complete media using 25 nM siRNA and Viromer Blue according to the manufacturer's instructions 1X (Lipocalyx). Knock-down gene expression was verified 48 h after transfection.

## Quantitative PCR in HEK293 cells

Total cellular RNA was isolated using RNeasy Kit according to the manufacturer's instructions (Qiagen) and quantified using a NanoDrop spectrophotometer (ND-1000; Thermo Fisher Scien- tific). cDNA was obtained using a RevertAid H Minus First Strand cDNA Synthesis Kit with random hexamers. Quantitative PCR (qPCR) was performed with a CFX Connect ther-mal cycler (Bio-Rad) using HOT FIREpol EvaGreen qPCR Super- mix (Solis Biodyne) and standard primers. Samples were measured as triplicates. The relative expression of each gene was calculated according to the delta-deltaCT method[87]. Expression of the housekeeping gene PPIA was used to normalize for variations in RNA input. PPIA and AdipoR2 primers were: AdipoR2 forward, TCATCTGTGTGCTGGGCATT; Adipo2 reverse, CTATCTGCCCT ATGGTGGCG; PPIA forward, GTCTCCTTTGAGCTGTTTGCAG; PPIA reverse, GGACA AGATGCCAGGACCCC.

## Western blot for HEK293 cells

Cellular proteins were extracted using a lysis buffer (1% Nonidet P-40, 0.1% SDS, 10% glycerol, 1% sodium deoxycholate, 1 mM DTT, 1 mM EDTA, 100 mM HEPES, 100 mM KCl) contain-ing Halt Protease Inhibitor Cocktail (1X; Pierce) on ice for 10 min. Upon lysis completion, cell lysates were centrifuged at 13 000 rpm for 10 min at 4˚C. The soluble fraction was kept for fur-ther analysis, and the protein sample concentration was quantified using the BCA protein assay kit (Pierce) according to the manufacturer's instructions. Twenty micrograms of protein were mixed with Laemmli sample loading buffer (Bio-Rad), heated to 37˚C for 10 min, and loaded in 4% to 15% gradient precast SDS gels (Bio-Rad). After electrophoresis, the proteins were transferred to nitrocellulose membranes using Trans-Blot Turbo Transfer Packs and a Trans-Blot Turbo apparatus/predefined mixed-MW program (Bio-Rad). Blots were blocked with 5% nonfat dry milk in PBS-T for 1 h at room temperature. Blots were incubated with pri-mary antibodies overnight at 4˚C: rabbit monoclonal anti-HA antibody (C29F4; Cell signaling Technology) 1:5000 dilution, mouse monoclonal anti-FLAG antibody (F3165; SIGMA) 1:3000 dilution, mouse monoclonal anti-VSV antibody (V5507; SIGMA) 1:3000 and rabbit anti-GAPDH antibody (14C10; Cell Signaling Technology). Blots were then washed with PBS-T and incubated with either swine anti-rabbit HRP (1:3000, Dako) or goat anti-mouse HRP (1:3000, Dako)) and washed again with PBS-T. Detection of the hybridized antibody was per-formed using an ECL detection kit (Immobilon Western; Millipore), and the signal was visual-ized with a digital camera (VersaDoc; Bio-Rad).

**Palmitic acid treatment of HEK293 cells.** Palmitic acid was dissolved in sterile DMSO (Sigma-Aldrich) and then mixed with fatty acid-free BSA (Sigma-Aldrich) in serum-free medium for 15 min at room temperature. The molecular ratio of BSA to palmitic acid was 1:5.3 when PA 400 µM and 1:2.65 when PA was 200 µM. Cells were then cultivated in serum-free media containing palmitic acid for 6 h prior to analysis.

Statistics. Unless otherwise stated, means and standard error of the means are presented, and *t*-tests were used to identify significant differences. For the tail tip defect, significant differences were determined using *Z-tests*. All experiments were repeated several times with similar results. Asterisks are used in the figures to indicate various degrees of significance, where *: $p < 0.05$; **: $p < 0.01$; and ***: $p < 0.001$.

## Supporting information

**S1 Fig. The *gof paqr-1(et52)* allele suppresses most or all *paqr-2* mutant phenotypes, and an R109A substitution also acts as a *gof paqr-1* allele.** Several phenotypes were scored for worms of the indicated genotypes. **(A)** Length of worms placed as L1s on NGM media and measured after 144 hours cultivation at 15˚C; **(B-C)** tail tip phenotype scored on 1-day old adults; **(D)** Length of worms of the indicated genotypes and placed as L1s on NGM media and measured after 144 hours cultivation at 15˚C. Note that both *paqr*-2 mutant transgenic lines carrying the PAQR-1(R109A) expression construct are glucose and cold tolerant, and have normal tails. **(E)** Tail tip phenotype scored on 1-day old adults. **(F)** pharyngeal pumping rate; **(G)** brood size; **(H)** life span; **(I)** defecation rate; **(J)** locomotion rate. Most *paqr*-2 mutant phenotypes were suppressed by the *paqr-1(et52)* gain-of-function mutation, and unaffected or worsened by the *paqr-1(tm3262)* loss-of-function mutation. The dashed line in D indicates the approximate size of the L1 larvae at the start of the experiment. Significant differences compared to the *paqr-2* genotype are indicated where: * $p < 0.05$, **$p < 0.01$ and ***$p < 0.001$ (ns: not significant).
(TIFF)

**S2 Fig. The *paqr-1(et52)* allele suppresses several membrane-related phenotypes of the *paqr-2* mutant.** **(A)** Photographs of worms of the indicated genotypes spotted as L1s then cultivated for 72 hours on NGM plates or on plates containing 20 mM glucose, or 144 hours on NGM plates incubated at 15˚C. **(B)** The *paqr-1(et52)* alleles suppresses the tail tip defect of the *paqr-2* mutant. **(C-D)** The *paqr-1(et52)* mutation suppresses the excess MUFA and depletion of MUFAs in the PEs of *paqr-2* mutant worms, especially when worms are grown on 20 mM glucose. **(E-H)** FRAP measurements showing that the *gof paqr-1(et52)* has no effect on the membrane fluidity of wild-type or *paqr-2* mutant worms grown on normal media. Significant differences compared to the *paqr-2* genotype are indicated where: * $p < 0.05$, **$p < 0.01$ and ***$p < 0.001$ (ns: not significant).
(TIFF)

**S3 Fig. Genetic requirements for *paqr-1(et52)* function.** **(A)** Length of worms placed as L1s on the indicated RNAi treatments and measured 72 hours later. **(B)** tail tip phenotype scored on 1-day old adults of the indicated genotypes. **(C)** Length of worms with the indicated genotypes placed as L1s on NGM media and measured after 144 hours cultivation at 15˚C. Note that *paqr-1(et52)* is unable to suppress the cold intolerance of the *paqr-2* mutant when *nhr-49* is also mutated. **(D)** Tail tip phenotype scored on 1-day old adults of the indicated genotypes. Note that *paqr-1(et52)* is able to suppress the cold tail tip defect of the *paqr-2* mutant even when *nhr-49* or *iglr-2* are also mutated. The dashed lines in A and C indicate the approximate size of the L1 larvae at the start of the experiment. Significant differences compared to the *paqr-2* genotype are indicated where: * $p < 0.05$, **$p < 0.01$ and ***$p < 0.001$ (ns: not significant).
(TIFF)

**S4 Fig. BiFC shows that PAQR-2, but not PAQR-1, interacts with IGLR-2. (A)** Schematic structures of IGLR-2 fused to VC155 and PAQR-2 fused to the VN173 fragment. BiFC could occur over time if the interaction between the two proteins allows reconstitution of a full and

fluorescent VENUS YFP. **(B)** Visualization of the BiFC signal on the plasma membrane of intestinal cells (arrowheads). **(C)** Schematic structures of IGLR-2 fused to VC155 and PAQR-1 (wild type or R109C variant) fused to the VN173 fragment. BiFC would not occurs over time if no interaction occurs between the two proteins. **(D and E)** No BiFC signal was detected on the plasma membrane when using either wild-type PAQR-1 or PAQR-1(R109C) as a possible partner for IGLR-2; the diffuse background signal is due to autofluorescence in the intestine. (TIFF)

**S5 Fig. Domain swapping experiments indicate that the intracellular domains of PAQR-2 and IGLR-2 are likely regulatory.** The cartoon representation of the various constructs used in this structure-function study are as in Fig 4A, and A/B indicate two separate transgenic lines for each construct in the indicated genetic background. **(A, C, E** and **G)** Length of worms placed as L1s on NGM media and measured after 72 hours of cultivation on NGM plates or plates containing 20 mM glucose. **(B, D, F** and **H)** Tail tip phenotype scored on 1-day old adults. The dashed lines in A, C and E indicate the approximate size of the L1 larvae at the start of the experiment. Significant differences compared to the *paqr-2* genotype are indicated where: $^*$ p<0.05, $^{**}$p<0.01 and $^{***}$p<0.001 (ns: not significant). (TIFF)

**S1 Lipidomics. Original lipidomics data.** The values are expressed in %mol for each listed fatty acid type. (XLSX)

## Acknowledgments

We acknowledge the Centre for Cellular Imaging at the University of Gothenburg and the National Microscopy Infrastructure, NMI (VR-RFI 2016–00968) for assistance in microscopy. We also thank Matilda Colm, Lisa Westlund for helping with phenotypic characterization of several *C. elegans* strains.

## Author Contributions

**Conceptualization:** Kiran Busayavalasa, Mario Ruiz, Ranjan Devkota, Emma Svensk, Jan Borén, Marc Pilon.

**Data curation:** Kiran Busayavalasa, Mario Ruiz, Ranjan Devkota, Marcus Ståhlman, Marc Pilon.

**Formal analysis:** Kiran Busayavalasa, Mario Ruiz, Ranjan Devkota, Marcus Ståhlman, Emma Svensk, Marc Pilon.

**Funding acquisition:** Marc Pilon.

**Investigation:** Kiran Busayavalasa, Mario Ruiz, Ranjan Devkota, Marcus Ståhlman, Rakesh Bodhicharla, Emma Svensk.

**Project administration:** Marc Pilon.

**Resources:** Nils-Olov Hermansson.

**Supervision:** Jan Borén, Marc Pilon.

**Visualization:** Kiran Busayavalasa, Mario Ruiz, Ranjan Devkota, Marc Pilon.

**Writing – original draft:** Marc Pilon.

**Writing – review & editing:** Kiran Busayavalasa, Mario Ruiz, Ranjan Devkota, Jan Borén, Marc Pilon.

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
