## [Decision Letter · Decision Letter 0]

13 Apr 2020

Dear Dr Pilon,

Thank you very much for submitting your Research Article entitled 'LEVERAGING A GAIN-OF-FUNCTION ALLELE OF C. ELEGANS PAQR-1 TO ELUCIDATE MEMBRANE HOMEOSTASIS BY PAQR PROTEINS' to PLOS Genetics. Your manuscript was fully evaluated at the editorial level and by independent peer reviewers. The reviewers appreciated the attention to an important problem, but raised some substantial concerns about the current manuscript. Based on the reviews, we will not be able to accept this version of the manuscript, but we would be willing to review again a much-revised version. We cannot, of course, promise publication at that time.

As you will see in their comments, the reviewers have indicated that certain claims should either be better supported by data or toned down, better documentation of the expression pattern of paqr-1, and experiments that could bolster the proposed interaction of PAQR-1 and IGLR-2.  Should you decide to revise the manuscript for further consideration here, your revisions should address the specific points made by each reviewer. We will also require a detailed list of your responses to the review comments and a description of the changes you have made in the manuscript.

If you decide to revise the manuscript for further consideration at PLOS Genetics, please aim to resubmit within the next 60 days, unless it will take extra time to address the concerns of the reviewers, in which case we would appreciate an expected resubmission date by email to plosgenetics@plos.org.

[LINK]

We are sorry that we cannot be more positive about your manuscript at this stage. Please do not hesitate to contact us if you have any concerns or questions.

Yours sincerely,

Kaveh Ashrafi

Associate Editor

PLOS Genetics

Gregory P. Copenhaver

Editor-in-Chief

PLOS Genetics

Reviewer's Responses to Questions

**Comments to the Authors:**

Reviewer #1: PAQR-1 and PAQR-2 are members of a conserved family of transmembrane hydrolases. Previously the Pilon lab demonstrated that PAQR-2 and IGLR-1 form a complex that is required to regulate membrane fluidity in response to cold temperatures and increased saturated fats that accumulate upon glucose feeding. In this study, the authors add to the understanding of this regulatory complex by showing that a PAQR-2 homolog, called PAQR-1, also contributes to the regulation of membrane fluidity. Their findings emphasize the N terminal cytoplasmic domain, which appears to have a regulatory function. Interestingly, the phenotype of a loss of function paqr-1 mutation is not apparent in lab-grown C. elegans, but the gain of function R109C allele is able to suppress the paqr-2 cold sensitivity and glucose sensitivity phenotypes. For the most part, I found this paper to be well written, the experiments rigorous and well executed, and the findings to be of importance to the field. The conservation with mammalian proteins, together with some experimental data using HEK293 cells demonstrates the importance of these findings to human physiology and medicine.

My concerns mainly deal with writing and presentation.

1. In the Introduction, the authors write (line 94), “…we performed a forward genetic screen for enhancers of the ability of mdt-15(et14) to suppress the SFA intolerance of the paqr-2(tm3410) null mutant..” However, after that, these mutants are always referred to as suppressors, including in the materials and methods. I understand that the mutants are suppressors of the paqr-2 phenotypes, as is mdt-15(et14), but referring to them as enhancers is confusing here. Please clarify the writing.

2. On line 162, this latter part of this sentence was confusing. “In contrast, the loss-of-function allele paqr-1(tm3262) has no paqr-2 suppressor effect or indeed significantly worsens paqr-2 mutant phenotypes such as poor growth with/without glucose.”  

Do they mean “…has no paqr-2 suppressor effect, nor does it significantly worsen the paqr-2 mutant phenotypes….” Or do they mean “…has no paqr-2 suppressor effect, but it significantly worsens the paqr-2 mutant phenotypes...” Looking at the data, trying to figure this out, my interpretation is that the double mutant showed only slight changes, but these were different depending on the phenotype observed:

Figure 2A, double mutant was shorter when grown on NGM, but same on glucose

Figure 2B, double mutant showed no change from paqr-2 mutant

Figure 2F-6, double mutant showed more SAT FA and less PUFA

Figure S2D, double mutant showed slight increase in MUFAs compared to paqr-2 mutant.

Figure S1F, double mutant lower pumping rate

Figure S1G, double mutant lower brood size

Figure S1H, double mutant Longer lifespan

Figure S1J, double mutant less movement

Therefore, in two cases the phenotype of the double mutant was the same, in six cases the phenotype of the double was worse than paqr-2 alone, and in two cases (the increased MUFAs and the longer lifespan) the phenotype of the double improved. However, in all cases the changes seem very small, even if they are statistically significant (except lifespan, that one improved a lot, although I’m concerned because the N2 lifespan looks too short compared to other studies). But the changes in fatty acid content are very small, so even if they are significant, they don’t seem large enough to be very important.

So maybe the better wording would be something like: . “In contrast, the loss-of-function allele paqr-1(tm3262) has no paqr-2 suppressor effect and sometimes slightly worsens paqr-2 mutant phenotypes such as …. (be precise here, only mention the phenotypes that are actually worse, and also be upfront about the changes that are very slight, such as the FA composition changes).”  

3. In the Abstract, end of Introduction, Results, and Discussion, the authors refer to the “downstream effectors”, with sentences like: “PAQR-1 does not require IGLR-2 but likely competes with PAQR-2 for downstream effectors”. This made me anticipate that downstream proteins interacting with the PAQR-2 complex were identified (perhaps a substrate?). However, as I read the paper I realized that by “downstream effectors” they meant the first steps of the fatty acid desaturation pathway (fat-5,6,7), which are regulated by sbp-1, nhr-49, and mdt-1. That these genes are required for increased fatty acid desaturation downstream of paqr-2 has been shown previously. This was disappointing, because using the term “downstream effectors” set the reader up for expecting a new discovery. Furthermore, the experiments to show that they are required for paqr-1 (GOF) suppression in Figure 3 were quite weak. RNAi was used, and the exact clones used were not described (Ahringer library?). The authors admitted that the bacterial host for RNAi, HT115, is quite a bad diet for paqr-2 and paqr-2/paqr-1(gof) mutants. In fact, the paqr-1(gof) only very slightly suppresses the paqr-2 under these conditions. Thus, the evidence is a minor change in this severe phenotype, making it slightly worse when the delta-9 desaturases and their regulators are depleted by RNAi. To be more convincing, the researchers need to find a way to do this outside of the HT115 background (some people have developed RNAi in OP50 background, or else they could make one or two triple mutants). RNAi of fat-5, fat-6, and fat-7 is tricky anyway, the Ahringer RNAi clones almost certainly knock down the other paralogs, especially fat-6 and fat-7, which have a very high degree of sequence similarity. In summary, the evidence that paqr-1 and paqr-2 “compete” for “downstream effectors” is very weak.

4. In Figure 6, the model is nice and it implies that the PAQR-1 and PAQR-2 are competing for the substrate of the hydrolase. Any speculation as to the substrate? This model emphasizes that the two proteins are not competing for effectors such as fatty acid desaturases or regulatory transcription factors, but for a substrate that might be involved in signaling, confirming the issues with evidence for “competing for downstream effectors” brought up in comment #3.

Reviewer #2: In this manuscript, Busalayava et al made a genetic analysis of membrane fluidity regulation in C. elegans by the PAQR-1 and PAQR2 paralogs. They previously reported that PAQR2 acts together with IGLR-2. C. elegans mutants in the genes coding for these proteins displayed several genotypes due to a lack of membrane fluidity optimization. Nevertheless, mutants in PAQR-1 did not display a similar phenotype that paqr-2 or IGLR-2 single mutant, suggestive that PAQR-1 was a silent protein. In this manuscript the authors the authors report the isolation and characterization of a gain of function (gof) allele of paqr-1 (paqr-1 et 52). Through a careful genetic analysis of this gof allele and domain swapping with paqr-2 they conclude that : the TM domain of PAQR-2 interacts with IGLR-2 and that the divergent N- terminal cytoplasmic domain of both, PAQR-1 and PAQR-2 protein likely regulates the access of a yet unknown substrate to the catalytic site of these protein.

This is a very interesting and solid manuscript, well written and illustrated, that present important advances in the mechanism of PAQR type proteins to achieve regulation of membrane homeostasis.

I just have minor questions:

One of the conclusions obtained with domain swapping is that the TM domains of PQR-2 dictate the requirement of IGLR-2 for activity, while genetic analysis shows that PAQR-1 is independent of IGLR-2. This is quite surprising since the TM domains of PAQR-1 and PAQR-2 are highly homologous! Do they test if a chimera of the TM domains of PAQR-1 and the entire N-terminal domain of PAQR-2 (rather that the N terminus of PAQR-1 (R109C)) produces an protein that requiring IGLR-2 ?

Do they authors have any data of bi-fluorescence assay between IGLR-2 and PAQR-1?

What is the evidence that PAQR-1 acts constitutively? (Rows 294-295). The genetic data show that this protein is inactive.

Is there evidence that AdipoR1 and AdipoR2 interacts with a putative IGLR2?

In materials and methods are described lipidomic assays that are not mentioned in the text.

Line 286 Discussion . What does mean “what portion was included seem…”?

Reviewer #3: The manuscript by Busayavalasa et al reported the identification of a new gain-of-function allele of paqr-1. Worms bearing the paqr-1(gf) allele suppressed the phenotypes caused by loss of paqr-2 function. These phenotypes include cold intolerance, a reduction in body length, tail morphology defects and intolerance to 20mM glucose. The last phenotype was previously shown by the authors to induce saturated fatty acid accumulation in E. coli, which when consumed by worms, supposedly increased membrane rigidity that was incompatible with the loss of paqr-2 function. In contrast, how did the loss of paqr-2 function reduce the body length and affect tail morphology remained mysterious, despite a series of papers from the same lab. The lack of clarity in the casual relationship between paqr-2(lf) and the phenotypes scored made the interpretation of the effect of the new paqr-1(gf) allele difficult.

The expression pattern of PAQR-1 was described mostly in words instead of being shown (Figure 1D). No attempt was made to demonstrate the site of action of PAQR-1. Although the authors claimed that “paqr-1(et52) suppresses all paqr-2 mutant phenotypes” (Line 149), the results on body length when worms were grown on glucose containing plates suggested otherwise (Figure 3B). The claim that “the intolerance to SFA, which is a harder challenge for the paqr-2 mutant” (Line 135) was also hard to understand. In the absence of a concrete molecular explanation, it is impossible to interpret the way different chimeric PAQR-1/PAQR-2 proteins functioned, in transgenic worms that were studied in various assays (Figure 4 and Figure S4). Therefore, the assignment of specific functions to the N-terminal cytosolic domain and the transmembrane domain of PAQR-2 and PAQR-1 was premature.

The genetic interaction studies on paqr-2(lf), iglr-2(lf) and paqr-1(gf) was insufficient to formulate the model proposed in Figure 6. The authors have previously used BiFC assays to demonstrate the interaction between PAQR-2 and IGLR-2. Similar molecular interaction studies are essential to place PAQR-1 in the model. The speculation on the role of the R109C gain of function mutation in PAQR-1 was also tenuous. The authors ruled out the possibility of aberrant disulfide bridges involving the extra cysteine residue. Instead, they proposed that the arginine residue was crucial to hold the N-terminal hypothetical regulatory domain in position, according to the model in Figure 6. If that is the case, the deletion of the entire N-terminal domain of PAQR-1 should provide the strongest suppression of paqr-2(lf) mutant phenotypes.

The results on the effects of overexpressing mammalian AdipoR1 or AdipoR2 in HEK293 cells were impossible to interpret. This is because the authors did not demonstrate if PAQR-1 and PAQR-2 are functional homologs of AdipoR1 and AdipoR2, respectively. Nor did they show if the expression of PAQR-1 in mammalian cells would trigger the same effect on membrane fluidity. The similarity in primary amino acid sequence is not a guarantee of functional similarity.

**Have all data underlying the figures and results presented in the manuscript been provided?**

Reviewer #1: Yes

Reviewer #2: Yes

Reviewer #3: None

PLOS authors have the option to publish the peer review history of their article (what does this mean?). If published, this will include your full peer review and any attached files.

Reviewer #1: No

Reviewer #2: No

Reviewer #3: No

---

## [Decision Letter · Decision Letter 1]

1 Jul 2020

Dear Dr. Pilon,

We are pleased to inform you that your manuscript entitled "LEVERAGING A GAIN-OF-FUNCTION ALLELE OF C. ELEGANS PAQR-1 TO ELUCIDATE MEMBRANE HOMEOSTASIS BY PAQR PROTEINS" has been editorially accepted for publication in PLOS Genetics. Congratulations!

Please note that Reviewer #3 has a handful of helpful textual suggestions (see below) that you should consider as you prepare the final version of the manuscript for the production team (the editorial team will not need to re-evaluate).

Yours sincerely,

Kaveh Ashrafi

Associate Editor

PLOS Genetics

Gregory P. Copenhaver

Editor-in-Chief

PLOS Genetics

Comments from the reviewers (if applicable):

Reviewer's Responses to Questions

**Comments to the Authors:**

Reviewer #1: The authors have addressed my concerns from the first review, and I don't have any further concerns.

Reviewer #2: The authors have made new experiments and text modifications that satisfactorily answered the comments I did in my previous review

Reviewer #3: The authors addressed most issues raised previously. Specific changes, detailed below, should be considered further.

1. Line 44. The authors stated a specific model that was not sufficiently substantiated in the current manuscript: “the divergent N-terminal cytoplasmic domains of the PAQR-1 and PAQR-2 proteins likely regulates access to the catalytic site of these proteins, as with the “ball-and-chain” mechanism found in certain voltage-gated channels.” First, has the catalytic site been defined? Second, “ball-and-chain” invokes a globular domain plus a flexible linker region, which can be demonstrated by partial proteolytic cleavage. This statement is fine in the Discussion section, but perhaps too speculative in the Abstract.

2. Line 112. The following statement “which suggests that the presence of paqr-2 inhibits paqr-1(et52) only when iglr-2 is absent” is ambiguous. If one interprets the sentence literally, one can re-write the sentence as: the paqr-2 gene inhibits a dominant allele of paqr-1, when the iglr-2 gene is mutated. This is confusing. Re-writing the sentence from the point of view of proteins would make it easier to read.

3. Line 179. Should nhr-49 be written as NHR-49? Same for SBP-1 and MDT-15 in the following two lines.

4. Line 194. Is there an alternative interpretation? That the paqr-1(et52) allele could not suppress all phenotypes that are caused by the loss of sbp-1 function, some of which are unrelated to PAQR-2 function.

5. Line 197. Should the sentence be written from the perspective of PAQR-2 and IGLR-2 proteins?

6. Line 273. Did the authors imply that AdipoR1 is orthologous to PAQR-1 and AdipoR2 is orthologous to PAQR-2? In this case, does the N-terminal domain of AdipoR1 share a high degree of similarity with PAQR-1 and not PAQR-2? Comments on the N-terminal domain of AdipoR1 and AdipoR2 would be helpful for readers.

**Have all data underlying the figures and results presented in the manuscript been provided?**

Reviewer #1: Yes

Reviewer #2: Yes

Reviewer #3: Yes

PLOS authors have the option to publish the peer review history of their article (what does this mean?). If published, this will include your full peer review and any attached files.

Reviewer #1: No

Reviewer #2: No

Reviewer #3: No

**Data Deposition**

http://datadryad.org/submit?journalID=pgenetics&manu=PGENETICS-D-20-00388R1

**Press Queries**

---

## [Editor Report · Acceptance letter]

28 Jul 2020

PGENETICS-D-20-00388R1 

Leveraging a gain-of-function allele of Caenorhabditis elegans paqr-1 to elucidate membrane homeostasis by PAQR proteins 

Dear Dr Pilon, 

We are pleased to inform you that your manuscript entitled "Leveraging a gain-of-function allele of Caenorhabditis elegans paqr-1 to elucidate membrane homeostasis by PAQR proteins " has been formally accepted for publication in PLOS Genetics! Your manuscript is now with our production department and you will be notified of the publication date in due course.

With kind regards,

Matt Lyles

PLOS Genetics

On behalf of:
